# A new method to detect and classify polar stratospheric NAT clouds derived from radiative transfer simulations and its first application to airborne IR limb emission observations

Christoph Kalicinsky[1], Sabine Griessbach[2], and Reinhold Spang[3]

[1]Institute for Atmospheric and Environmental Research, University of Wuppertal, Germany
[2]Forschungszentrum Jülich, Jülich Supercomputing Centre, JSC, Jülich, Germany
[3]Forschungszentrum Jülich, Institut für Energie und Klimaforschung, Stratosphäre, IEK-7, Jülich, Germany

**Correspondence:** C. Kalicinsky (kalicins@uni-wuppertal.de)

**Abstract.** Polar stratospheric clouds (PSCs) play an important role for the spatial and temporal evolution of trace gases inside the polar vortex due to different processes, such as chlorine activation and NOy redistribution. As there are still uncertainties in the representation of PSCs in model simulations, detailed observations of PSCs and information on their type, nitric acid trihydrate (NAT), supercooled ternary solution (STS), and ice, are desirable.

The measurements inside PSCs by the airborne infrared limb sounder CRISTA-NF (CRyogenic Infrared Spectrometers and Telescope for the Atmosphere – New Frontiers) during the RECONCILE (Reconciliation of essential process parameters for an enhanced predictability of Arctic stratospheric ozone loss and its climate interactions) aircraft campaign showed a spectral peak at about $816\,\mathrm{cm^{-1}}$. This peak is shifted compared to the known peak at about $820\,\mathrm{cm^{-1}}$, which is known to be caused by emission of radiation by small NAT particles. To investigate the reason for this spectral difference we performed a large set of radiative transfer simulations of infrared limb emission spectra in the presence of various PSCs (NAT, STS, ice, and mixtures) for the airborne viewing geometry of CRISTA-NF. NAT particles can cause different spectral features in the region $810 - 820\,\mathrm{cm^{-1}}$. The simulation results show that the appearance of the feature changes with increasing median radius of the NAT particle size distribution from a peak at $820\,\mathrm{cm^{-1}}$ to a shifted peak and, finally, to a step-like feature in the spectrum, caused by the increasing contribution of scattering to the total extinction. Based on the appearance of the spectral feature we defined different colour indices to detect PSCs containing NAT particles and to subgroup them into three size regimes: small NAT ($\leq$ 1.0 µm), medium NAT (1.5 – 4.0 µm), and large NAT ($\geq$ 3.5 µm) under the assumption of spherical particles. Furthermore, we developed a method to detect the bottom altitude of a cloud by using the cloud index (CI), a colour ratio indicating the optical thickness, and the vertical gradient of the CI. Finally, we applied the methods to observations of the CRISTA-NF instrument during one local flight of the RECONCILE aircraft campaign and found STS and medium sized NAT.

# 1 Introduction

Polar stratospheric clouds (PSCs) form inside the cold polar vortices in both hemispheres in winter. They have a major influence on the ozone chemistry and thus the ozone depletion in the stratosphere (Solomon, 1999). PSCs are classified into three different types: supercooled ternary solution (STS) droplets, nitric acid trihydrate (NAT), and ice particles (e.g. Lowe and MacKenzie, 2008). The formation and existence of these different types is largely temperature dependent. Solid ice particles can only exist below the frost point $T_{\mathrm{frost}} \approx 188$ K, whereas NAT particles are thermodynamically stable at temperatures below $T_{\mathrm{NAT}} \approx 195$ K. The liquid STS droplets form from binary $H_2SO_4$-$H_2O$ droplets at temperatures below the dew point of $HNO_3$ $T_{\mathrm{dew}} \approx 192$ K by the uptake of $HNO_3$ (see e.g. Peter and Grooß, 2012, and references therein).

PSCs directly and indirectly influence the spatial distribution of trace gases relevant for ozone depletion in different ways. Due to heterogenous reactions on the cold particle surfaces chlorine is activated from its reservoir species, mainly HCl and $ClONO_2$, and the chlorine radicals catalytically destroy ozone (e.g. Solomon, 1999). NAT particles can grow to larger sizes, which then leads to a sedimentation of the particles and thus to a permanent removal of $HNO_3$ from the stratosphere (denitrification) (e.g. Fahey et al., 2001; Molleker et al., 2014). This denitrification slows down the deactivation of chlorine and thus enhances the ozone loss (e.g. Waibel et al., 1999; Peter and Grooß, 2012).

Because of these different processes and due to the fact that many models rely on rather simple parametrisations, the simulation of PSCs and related processes is incomplete and partly accompanied with larger uncertainties. Especially, chemistry climate models (CCMs) that are used to asses polar stratospheric ozone loss (e.g Eyring et al., 2013) often use rather simple schemes to represent PSCs in the model simulations. Such simplifications may lead to a heterogeneous chemistry dominated by NAT, but it is known that heterogeneous chemistry on STS and cold binary aerosol particles probably dominates the chlorine activation (e.g Solomon, 1999; Drdla and Müller, 2012; Kirner at al., 2015). Additionally, no comprehensive microphysical models are typically used to describe the evolution of PSCs over the winter. Mesoscale temperature variations that are known to play an important role for the formation of PSCs (Carslaw et al., 1998; Dörnbrack et al., 2002; Engel et al., 2013; Hoffmann et al., 2017) are also missing in current state of the art CCMs (Orr et al., 2015).

Assumptions on the occurence of different PSC types typically have only limited impact on many aspects of ozone loss, as, for example, liquid PSC particles are sufficient to simulate nearly all ozone loss (Wohltmann et al., 2013; Kirner at al., 2015; Solomon et al., 2015). However, there are also situations where the PSC type is crucial. For example, which PSC type is present at top of ozone loss region is important (Kirner at al., 2015) and during the initial activation in PSCs covering only a small part of the vortex the type plays also an important role (Wegner et al., 2012). Furthermore, the heterogenous reaction rates on PSCs strongly depend on temperature but also on the PSC type (e.g. Drdla and Müller, 2012; Wegner et al., 2012). Here, especially for NAT the reaction rates show rather large uncertainties (Carslaw et al., 1997; Wegner et al., 2012), which highlights the importance of observing the PSC type.

In summary, information on the compostion of the PSCs is very important, but measurements are limited. Typical PSC measurement techniques are in-situ particle measurements and lidar observations (e.g. Molleker et al., 2014; Achtert and Tesche, 2014; Pitts et al., 2018). But beside these measurement techniques an infrared limb emission sounder builds a good basis for

such kind of studies (e.g. Spang and Remedios, 2003; Höpfner et al., 2006).

In addition to the derivation of volume mixing ratios of several trace gases, infrared limb emission sounders such as CRISTA (CRyogenic Infrared Spectrometers and Telescopes for the Atmosphere; Offermann et al., 1999; Grossmann et al., 2002), CRISTA-NF (CRyogenic Infrared Spectrometers and Telescope for the Atmosphere – New Frontiers; Kullmann et al., 2004), and MIPAS-Envisat (Michelson Interferometer for Passive Atmospheric Sounding - Envisat; Fischer et al., 2008) are well

suited to detect clouds of different composition. Spang et al. (2001) established a simple and effective way for cloud detection based on the radiance ratio of two specific spectral regions. The first one is dominated by $CO_2$ molecular line emissions (around $792 \ cm^{-1}$) and the second one by the broader continuum like aerosol emissions (around $833 \ cm^{-1}$). This detection method was then succesfully used in various studies and for different satellite and airborne instruments (e.g. Spang and Remedios, 2003; Spang et al., 2002, 2005, 2008, 2015; Höpfner et al., 2006; Kalicinsky et al., 2013). Furthermore, the infrared spectra

exhibit different spectral features or radiance behaviours in the presence of polar stratospheric clouds of different type. Spang and Remedios (2003) presented a sharp peak-like feature at around $820 \ cm^{-1}$ observed by the CRISTA instrument in the Antarctic winter for the first time. They attributed this feature to $HNO_3$-containing particles. Höpfner et al. (2006) showed that the feature can be best reproduced in simulations by using refractive index data for $\beta$-NAT particles (assuming relatively high number densities). The authors showed that the colour ratio method derived by Spang and Remedios (2003) that exploits the

peak-like feature for detection is able to identify PSCs containing small NAT particles (radii < 3 µm). By means of a large set of radiative transfer simulations Spang et al. (2012, 2016) developed a detection and discrimination method for PSCs. Besides the feature at $820 \ cm^{-1}$, the method also uses further spectral behaviours such as a radiance decrease from around $830 \ cm^{-1}$ towards larger wavenumbers (around $950 \ cm^{-1}$) that occurs in the case of ice to distinguish different PSC types. Spang et al. (2018) finally presented a climatology of the PSC composition for the whole MIPAS-Envisat observation time period (2002 –

2012) based on this method.

During the RECONCILE campaign (Reconciliation of essential process parameters for an enhanced predictability of Arctic stratospheric ozone loss and its climate interactions; von Hobe et al., 2013) a slightly different type of spectral feature was observed by the infrared limb emission sounder CRISTA-NF in the presence of PSCs. The campaign took place in Kiruna, Sweden, in January to March 2010 and the high-flying research aircraft M55-Geophysica carried out a number of flights with

a huge set of different instruments to study the polar vortex and related scientific topics such as PSCs, ozone chemistry, or mixing processes. CRISTA-NF was one of the operated instruments and the instrument is able to detect clouds and distinguish different types of PSCs. During the first five flights of the campaign (17.01. – 25.01.2010) CRISTA-NF detected PSCs around the aircraft. Because of the viewing geometry of an infrared limb sounder, where the horizontal distance of a tangent point to the aircraft is increasing with decreasing sampling altitude, CRISTA-NF typically observes air masses in a wider range around

the aircraft. The infrared spectra measured inside the clouds show a noticeable spectral feature at about $816 \ cm^{-1}$. This feature is slightly shifted towards smaller wavenumbers compared to the typical spectral feature at $820 \ cm^{-1}$, which is caused by very small NAT particles. Another appearance of the spectral feature caused by NAT particles, a step-like or shoulder-like behaviour of the radiance in the spectral region $810 – 820 \ cm^{-1}$, has been observed by the ballon-borne instrument MIPAS-B in January 2001 (Höpfner et al., 2002) and the airborne instrument MIPAS-STR during the ESSENCE campaign in December 2011

(Woiwode et al., 2016). Woiwode et al. (2016) analysed different particle modes with radii between 1 and 6 μm and suggested that highly aspherical medium/large sized (median radius 4.8 μm) NAT particles caused the change of the NAT signature to the step-like appearance. Woiwode et al. (2016) concluded that the particle shape and the scattered radiation from below can influence the feature. However, the appearance of the spectral feature seems to depend on the particle size distributions of the NAT particles, especially on the median radius. This motivated the following study of the relationship between the appearance of the feature (typical, shifted, and step-like) and the corresponding particle size distribution aiming at an improved detection method for PSCs containing NAT particles. Especially the possibility to detect larger NAT particles is important, because they play the major role for the denitrification of the stratosphere (Fahey et al., 2001; Molleker et al., 2014), and thus for the whole ozone chemistry (e.g. Waibel et al., 1999; Peter and Grooß, 2012). New insights and an improvement of the detection method is also interesting for measurements of other infrared limb emission sounder such as MIPAS-Envisat, where data for a long observation period (about 10 years) and both hemispheres are available. For this purpose a large variety of different PSCs with respect to the composition, spatial dimensions, and particle size distributions of the PSC particles was simulated and analysed. The paper is structured as follows. In Sect. 2 the CRISTA-NF instrument and the radiative transfer code are described and the setup used for the simulations is explained. The results of the simulations are presented and analysed in Sect. 3. The derived methods are applied to the CRISTA-NF measurements in Sect. 4. Finally, the main results are discussed in Sect. 5 and summarised in Sect. 6.

## 2 Methods and setup

The radiative transfer simulations were performed using the viewing geometry and spectral properties of the airborne infrared limb sounder CRISTA-NF. The background atmosphere used for the simulations is a polar winter atmosphere with conditions suitable for the formation of polar stratospheric clouds. The simulations themselves were performed using the Juelich Rapid Spectral Simulation Code (JURASSIC). This section describes the CRISTA-NF instrument, the JURASSIC radiative transfer code, the setup used for the simulations, and the different cloud scenarios that were investigated.

### 2.1 CRISTA-NF istrument

The airborne CRISTA-NF instrument is a successor of the satellite instrument CRISTA. CRISTA-NF measures the thermal emissions of the atmosphere in the mid-infrared region from 4 to 15 μm in an altitude range from flight altitude (up to 20 km) down to approximately 5 km. A detailed description of the design of the cryostat and the optical system is given by Kullmann et al. (2004). The calibration procedure and some improvements for the RECONCILE campaign are described by Schroeder et al. (2009) and Ungermann et al. (2012), respectively.

CRISTA-NF uses a Herschel telescope and a tiltable mirror to scan the atmosphere from flight altitude down to approximately 5 km. The incoming radiance is then spectrally dispersed by two Ebert-Fastie grating spectrometers (e.g., Fastie, 1991). These two spectrometers have different spectral resolving powers of $\frac{\lambda}{\Delta\lambda} \approx 1000$ and 500, respectively, and are therefore denoted as high resolution spectrometer (HRS) and low resolution spectrometer (LRS). In this study we focus on the spectral range that

is covered by the two channels LRS 5 ($850 - 965$ cm$^{-1}$) and LRS 6 ($775 - 865$ cm$^{-1}$). The vertical sampling of the LRS during one altitude scan is typically between 100 and 200 m with finer sampling at higher altitudes and a wider one at lower altitudes due to mounting and instrument conditions. The field of view (FOV) is very small with 3 arcmin (about 300 m at 10 km tangent height; Spang et al., 2008). The horizontal sampling along the flight track is about 12.5 to 15 km depending on the speed of the aircraft. The radiance is finally measured using liquid helium cooled semiconductor detectors (Si:Ga) that are operated at a temperature of 13 K. These low temperatures enable fast measurements of one spectrum in about one second.

## 2.2 Radiative transfer simulation code JURASSIC

For the simulations of the CRISTA-NF infrared spectra in the presence of PSCs we used the Juelich Rapid Spectral Simulation Code (JURASSIC) (Hoffmann, 2006). It is a fast radiative transfer model for the mid-infrared spectral region. It was used in numerous studies analysing infrared limb and nadir measurements, including MIPAS-Envisat (Hoffmann et al., 2005, 2008), CRISTA-NF (Hoffmann et al., 2009; Weigel et al., 2009), and the nadir sounder AIRS (Hoffmann and Alexander, 2009), and to simulate 2-D trace gas and temperature retrievals for a proposed new infrared limb instrument named PREMIER IRLS (PRocess Exploitation through Measurements of Infrared and millimetre-wave Emitted Radiation – InfraRed Limb Sounder; Preusse et al., 2009; Hoffmann and Riese, 2010). For fast radiative transfer calculations, JURASSIC applies a spectral averaging approach by using the emissivity growth approximation (EGA) (Gordley and Russel III, 1981; Marshall et al., 1994) and precalculated look-up tables. The look-up tables were calculated by the line-by-line model reference forward model (RFM; Dudhia, 2017) and take into account the spectral resolution of the instrument to be investigated.

JURASSIC has been compared to the line-by-line models RFM and KOPRA (Karlsruhe Optimized and Precise Radiative transfer Algorithm) for selected spectral windows and shows good agreement (Griessbach et al., 2013). JURASSIC was extended with a scattering module that allows for radiative transfer simulations including single scattering on aerosol and cloud particles (Griessbach et al., 2013). The optical properties of the particles (extinction coefficient, scattering coefficient, and phase function) required for the radiative transfer simulations with scattering, can either be calculated with an internal Mie code assuming spherical particles, or can be taken from databases for non-spherical particles. The scattering module was successfully used in different studies (Griessbach et al., 2014, 2016, 2020).

## 2.3 Simulation setup

The simulation setup can be divided into three parts: the instrument part, the atmosphere part, and the cloud scenarios. The instrument part includes the viewing geometry and the spectral properties of the airborne infrared limb emission sounder CRISTA-NF. In the atmosphere part we describe the background atmosphere that is used for the simulations. All different types of polar stratospheric clouds with respect to the position and thickness of the cloud as well as the composition (NAT, STS, ice) are summarised in the cloud scenario section.

### 2.3.1 Instrument properties

The two important spectral regions that are necessary to analyse infrared spectra with respect to polar stratospheric clouds are $785 - 840$ cm$^{-1}$, because of the cloud index (CI) and the NAT signature, and the region $940 - 965$ cm$^{-1}$, because of an ice signature. The spectral resolving power used in the simulations is $\frac{\lambda}{\Delta\lambda} = 536$ at a reference wavelength of 12.5 μm ($800$ cm$^{-1}$) (see Weigel, 2009). The spectral sampling of the CRISTA-NF measurements is about 0.0065 μm that corresponds to an average of $0.42$ cm$^{-1}$ for the wavelength range $785 - 840$ cm$^{-1}$ and $0.59$ cm$^{-1}$ for the region $940 - 965$ cm$^{-1}$. These values have been considered in the calculation of the look-up tables for JURASSIC. We used an observer altitude of 18.4 km, which is the maximum average flight altitude during the RECONCILE flights of interest. However, the average altitudes of all flights only differ by a few hundred meters (18.1 to 18.4 km). The observations were simulated in the tangent altitude range from observer altitude down to 10 km. The vertical sampling used for the simulations was 100 m. Tab. 1 summarises these properties.

### 2.3.2 Atmospheric setup

For the background atmosphere we used polar winter conditions to get representative simulations. Most information were taken from the MIPAS reference polar winter climatology by Remedios et al. (2007). For some constituents we made updates and here we focused on the winter 2009/2010, especially on the January 2010. There are two reasons for this choice: 1. A large variety of PSCs has been observed in the Arctic in this winter. 2. The CRISTA-NF observations of PSCs during RECONCILE took place in January 2010. The updates are summarised in the following part.

Two very important parameters of the atmosphere for the simulation of infrared spectra are temperature and $CO_2$ volume mixing ratios (VMR). In order to have a temperature profile that is representative for a situation where many different PSCs can occur we focused on the observation period of PSCs during the RECONCILE aircraft campaign (17 to 25 January 2010). The temperature profile was derived using ERA-Interim reanalysis data (Dee et al., 2011). An average profile in the region of the CRISTA-NF observations north of Kiruna ($67° - 78°$ N, $10° - 35°$ E) was used for the background atmosphere. Additonally, we used the corresponding pressure profile from ERA-Interim reanalysis data. The $CO_2$ VMR we derived from the reconstructed $CO_2$ product by Diallo et al. (2017) for January 2010. The profile is a zonal average in the latitude region of interest. The profiles were extended to larger altitudes following the slope of the climatology. As PAN (peroxyacetyl nitrate) is not included in the climatology, we took a mean profile derived from CRISTA-NF observations (see Ungermann et al. (2012) for retrieval description) in the region around Kiruna between end of January and begin of March. Unfortunately, there are no retrieval results for PAN during the PSCs flights available. However, the derived profile is in a good agreement with published MIPAS-Envisat observations for October to December 2003 in the corresponding latitude region (Glatthor et al., 2007) and also with the average profile in 2007/08 in the latitudinal band $60° - 90°$ (Pope et al., 2016). For CFC-11, CFC-113, HCFC-22, SF$_6$, and COF$_2$ we updated the climatological profiles to 2010 values by using information of tropospheric values (Bullister, 2011) as well as satellite observations by ACE-FTS (Boone et al., 2013) and in-situ observations carried out by the HAGAR instrument (Riediger et al., 2000; Werner et al., 2010) onboard the M55-Geophysica during RECONCILE.

In order to save computation time we restricted the number of trace gases to a minimum by just using those gases that have a

185 noticeable contribution to the total radiance in the two analysed spectral regions. The trace gases have been selected seperately for the two spectral regions. For the region 785–840 $\mathrm{cm}^{-1}$ we used 13 trace gases: $CO_2$, $HNO_3$, $ClONO_2$, $O_3$, $H_2O$, $HNO_4$, $CCl_4$, CFC-11, HCFC-22, CFC-113, PAN, ClO, $NO_2$. The simulations in the region 940–965 $\mathrm{cm}^{-1}$ included 9 trace gases: $CO_2$, $HNO_3$, $O_3$, $H_2O$, CFC-11, PAN, $SF_6$, $NH_3$, $COF_2$. A summary of the trace gases, their sources, and the spectral region in which they were considered is given in Tab. 2.

## 2.4 Cloud scenarios

Two parameters that were largely varied to investigate different situations are the position and the thickness of the PSCs. The PSC position was varied between a minimum cloud bottom height (CBH) of 13 km and a maximum top height of 30 km. For the thickness we used the values 0.5, 1.0, 2.0, 4.0, and 8.0 km. The bottom height is shifted in 1 km steps up to 20 km (slightly above flight altitude) and in 2 km steps above for each thickness value as long as the cloud top height (CTH) is lower or equal
195 to 30 km.

PSCs consisting of NAT particles are the most interesting ones for this study, because of their impact on the NOy redistribution. Thus, the largest part of the simulations were performed for this particle type. The two parameters varied for the NAT scenarios are the median radius of the particle size distribution and the number density of the particles. The different particle size distributions for all cases (also ice, STS) were described by a log-normal distribution

$$\frac{\mathrm{d}N}{\mathrm{d}r} = \frac{N_0}{\sqrt{2\pi}\ln(\sigma)r}e^{-\frac{(\ln(r)-\ln(\mu))^2}{2\ln^2(\sigma)}}, \tag{1}$$

where $r$ is the radius, $N_0$ is the number density, $\mu$ is the median radius, and $\sigma$ is the width. The width is constant at $\sigma = 1.35$ and we varied the median radius between 0.5 and 8.0 µm. For the calculation of the number densities and thus the particle size distributions we used different $HNO_3$ VMRs from 1 to 15 ppbv under the assumption that one $HNO_3$ molecule is converted to one NAT molecule. The calculations were done for typical conditions for the lower stratosphere with 193 K and 60 hPa.
We also simulated PSCs using bimodal NAT particle distributions. The median radius of the first mode varied from 0.5 to 2.5 µm and was combined with a second mode, where larger radii than in the first mode were used. The total $HNO_3$ VMR was always 10 ppbv and the ratios between first and second mode were 70/30, 50/50, and 30/70. Here, we restricted the simulations to two bottom altitudes at 13 and 17 km and three different thicknesses 1, 4, and 8 km.

The volume densities used for the STS and ice simulations range from 0.1 – 10.0 µm³/cm³ and from 0.1 – 100.0 µm³/cm³,
respectively. The median radii were varied between 0.1 and 1.0 µm for STS and between 1.0 and 10.0 µm for ice. In the case of STS we simulated three different mixtures of $H_2SO_4$/$HNO_3$ with wt% of 2/48, 25/25 and 48/2.

Finally, we simulated mixed NAT/STS clouds. Here, we also used only the bottom altitudes at 13 and 17 km and the different thicknesses 1, 4, and 8 km like in the case of bimodal NAT. Furthermore, we concentrated on the small and medium size NAT particles up to 3.5 µm and used three different $HNO_3$ VMRs of 5, 10, and 15 ppbv. We combined these NAT scenarios with
215 STS scenarios using wt% 2/48, volume densities of 5 and 10 µm³/cm³, and radii of 0.1, 0.3, and 1.0 µm.

The parameter ranges for all cloud scenarios are summarised in Tab. 3 including the cloud extinction range. The refractive indices for ice, NAT, and the STS mixtures were taken from Toon et al. (1994), Biermann (1998) with refinement in Höpfner

et al. (2006), and Biermann et al. (2000), respectively. The total number of scenarios was 16392 (NAT: 9240, STS: 3360, ice: 2352, and mixtures bimodal NAT and NAT/STS: 1440).

Considering the computational resources required to simulate this amount of scenarios we assumed single scattering. Neglecting multiple scattering introduces uncertainties. Based on the findings of Höpfner and Emde (2005) for our optical depth ranges (Tab. 3, extinction times cloud thickness) and the single scattering albedos of STS, NAT and ICE the uncertainty is mostly below 1%, 4%, and 4%, respectively. Only for a few scenarios of 8km thickness the uncertainty may reach up to 4.5% for STS and 20% for ice. In the cloud scenario setup we assumed homogeneous clouds. The horizontal extent of the CRISTA-NF line of sight inside the PSC can reach up to several hundred kilometres. In case of synoptic scale PSCs horizontal homogeneity is a sufficiently good approximation. PSCs of other origin, e.g. mountain wave induced PSCs, have a smaller horizontal extent, but are less frequent compared to synoptic scale PSCs.

## 3 Results of the simulations

The following section deals with the analyses of the simulation results. The analyses can be divided into three parts. 1. The shift of the NAT feature; 2. The detection of PSCs and the identification of the PSC types NAT and ice; 3. The determination of the bottom altitude of the clouds.

### 3.1 Shift of the NAT feature

The appearance of the spectral feature that is observed in infrared limb spectra in the presence of polar stratospheric clouds consisting of NAT particles in our study is unambiguously characterized by the median radius of the particle size distribution, as we kept the distribution width $\sigma$ constant. With increasing median radius the shape transforms from the well known pronounced peak at about $820 \, \text{cm}^{-1}$ to a peak, which is slightly shifted towards smaller wavenumbers, and, finally, to a step-like feature. Figure 1 illustrates this behaviour and the dependency of the appearance on the median radius. The spectra shown in Fig. 1 are scaled such that the radiance in the spectral range $832.0 – 834.0 \, \text{cm}^{-1}$ is equal to one for all spectra. The example spectrum for the smallest median radius of 0.5 μm (yellow colour) exhibits a clear pronounced peak at about $820 \, \text{cm}^{-1}$. For slightly larger median radii (1.0 – 3.5 μm) the peak shifts to smaller wavenumbers and becomes less pronounced (orange colours). When the median radius is even larger the spectral feature transforms to a step-like feature that shows a steep radiance decrease from about $811 \, \text{cm}^{-1}$ to $826 \, \text{cm}^{-1}$. The magnitude of this decrease largely diminishes with increasing median radius.

This behaviour and thus the dependency of the appearance of the feature on the median radius can be explained by the different contributions of emission/absorption and scattering to the total observed radiance enhancement caused by the PSCs. These contributions largely depend on the median radius of the particles. The real and the imaginary part of the refractive index of $\beta$-NAT are shown in Fig. 2 a) in black and red, respectively. The imaginary part illustrates the emission and absorption characteristic of the NAT particles whereas the real part shows the scattering behaviour. The imaginary part shows a distinct peak at about $820 \, \text{cm}^{-1}$. As the emission is the major contribution when only small particles are present the simulated spectra in case of small NAT only show this peak. This is illustrated in Fig. 2 b) and c), where the extinction and single scattering

albedo (SSA) are shown. The extinction shows a clear peak at about 820 cm$^{-1}$ and the SSA, which gives the contribution of the scattering to the total extinction, shows low values. With increasing median radius of the NAT particles the scattering becomes more and more important. This can be seen by the increase of the SSA, which for large particles then can exceed 0.5, i.e. the scattering increasingly accounts for more than half of the total extinction. As a consequence the peak shifts to smaller wavenumbers with increasing particle size until it transforms to a step-like signature (see Fig. 1 and Fig. 2 b)).

The spectral feature with all its versions in the region 810 – 820 cm$^{-1}$ is a unique signature that only occurs in the presence of NAT PSCs and will be used for the detection of NAT PSCs. Other PSCs consisting of STS or ice do not show such a feature as exemplarily shown in Fig. 1 for two examples with light blue (STS) and dark blue (ice) colours. In contrast to the other PSC types ice shows the largest relative difference between the region 832.0 – 834.0 cm$^{-1}$ and the second spectral range of our simulations (940 – 965 cm$^{-1}$). Only NAT PSCs consisting of particles with very small radii ($< 1.0\,\mu$m) can also achieve large

differences. This spectral behaviour in the case of ice will further be used to detect ice PSCs.

## 3.2 Detection of NAT

### 3.2.1 Unimodal pure NAT

The detection of clouds using infrared limb spectra typically uses the cloud index (CI) (e.g Spang et al., 2001, 2002, 2008). The CI is the radiance ratio between a spectral region dominated by $CO_2$ at around 792 cm$^{-1}$ and a second spectral region

dominated by aerosol or cloud particles at around 833 cm$^{-1}$. For the analysis of the airborne observations by CRISTA-NF we used the two windows 791.0 – 793.0 cm$^{-1}$ (micro window MW1) and 832.0 – 834.0 cm$^{-1}$ (MW2). Because of the different viewing geometries and instrument properties, the first window is smaller defined than that typically used for satellite observations (Spang et al., 2008). In cloud free conditions the CI value (MW1/MW2) is typically large (around 10). When clouds or larger aerosol loads are in the line of sight of the instrument the CI significantly drops to smaller values depending

on the optical thickness of the cloud. The MWs used for the CI and all following indices are marked with gray bars in Fig. 1. Additionally, all indices are summarised in Tab. 4.

The detection of NAT particles inside clouds is based on the characteristic spectral behaviour in the region 810 – 820 cm$^{-1}$. In former studies a NAT index, the radiance ratio between the spectral region of the typical NAT feature (819 – 821 cm$^{-1}$) and the region of the $CO_2$-peak (788 – 796 cm$^{-1}$), was introduced to detect PSCs containing small NAT particles (e.g Spang and

275 Remedios, 2003; Höpfner et al., 2006; Spang et al., 2012, 2016, 2018). Here, we define the NAT index-1 as the radiance ratio between the two regions 819 – 821 cm$^{-1}$ (MW3) and 791 – 793 cm$^{-1}$ (MW1) and use the same smaller window in the region of the $CO_2$-peak as for the CI. In a scatter plot of the NAT index-1 versus the CI spectra simulated for small NAT particles separate from spectra simulated for larger NAT particles and other types of PSCs (Fig. 3 a)). Note here, that there is sometimes an overlap of the data points for adjacent radii. This is also the case in the other scatter plots. Nearly all simulations with NAT

particles $< 3\,\mu$m lie above the region of the simulations for STS and ice clouds, which is marked by the solid red line. The separation line was determined solely by the simulation results for ice and STS to define an upper limit of these simulations for each possible CI value. Thus, NAT particles within this size range can be detected and discriminated using NAT index-1.

In order to detect PSCs containing larger NAT particles and to make the estimation of the size range of the particles more distinct we introduce two new NAT indices here. The NAT index-2, which is defined as the radiance ratio between $815 - 817 \, \mathrm{cm}^{-1}$ (MW4) and $791 - 793 \, \mathrm{cm}^{-1}$ (MW1), focuses on the shifted NAT feature that occurs for larger particles than that producing the non-shifted feature. Figure 3 b) shows the scatter plot of the NAT index-2 versus the CI. In contrast to the NAT index-1 now the results for PSCs with larger NAT particles (up to 4 µm) also lie above the simulations for STS and ice (red separation line). Additonally, for different particle median radii the distance to the separation line changes when going from NAT index-1 to index-2 in an opposite way. In the case of very small particles the distance becomes smaller as the spectral region used for the detection moves away from the center of the typical NAT peak. For larger particles the behaviour is opposite and the distance enlarges as the spectral region moves to the center of the shifted NAT peak. This opposite behaviour can be seen in Fig. 3 c) where the difference between NAT index-1 and index-2 is shown on the y-axis. It is obvious that the simulations for the two smallest radii (0.5 and 1.0 µm) behave opposite to the simulations for larger radii.

Lastly, we introduce NAT index-3, which is defined as the radiance ratio between $810 - 812 \, \mathrm{cm}^{-1}$ (MW5) and $825 - 827 \, \mathrm{cm}^{-1}$ (MW6), to detect PSCs containing even larger NAT particles. This ratio enables the discrimination of a step-like feature from a peak and a more or less constant radiance in the complete spectral range as it is the case for STS and ice. Figure 3 d) shows the scatter plot of NAT index-3 against the CI. The simulation results for small NAT particles ($\leq 1.0$ µm) have a NAT index-3 smaller than for the simulations results for STS and ice. A large part of the simulation results for larger NAT particles show a NAT index-3 that is larger than for the simulation results for STS and ice. By using this ratio also NAT particles with radii $> 4$ µm separate from the other simulation results and are consequently detectable.

In total the three different NAT indices enable the detection of PSCs containing NAT particles and allow for a classification of the NAT particles in different size regimes. The detection and discrimination of NAT can be divided into three different cases: Case-1 (small NAT): Detection of NAT using NAT index-1 and the difference NAT index-1 – index-2 is above the separation line; Case-2 (medium NAT): Detection of NAT using NAT index-2 and the difference NAT index-1 – index-2 is below the separation line; Case-3 (large NAT): No Detection of NAT using NAT index-1 and index-2 (both below separation line), but the NAT index-3 is above the separation line. Figure 4 shows the proportion between the spectra detected as NAT influenced and the cloud spectra (spectra below a certain CI threshold) in each size bin of the simulations (0.5 – 8.0 µm) colour coded for the three different cases. In panel a) of Fig. 4 only observations with a CI below 5.0 are taken into account. Obviously, in case-1 (yellow colour) only NAT particles with a median radius of 0.5 or 1.0 µm are detected. The detection results in case-2 (orange colour) go from 1.5 µm up to 4.0 µm and the results in case-3 (yellow colour) all show median radii larger than or equal to 2.5 µm, whereby the radius 2.5 µm only occurs in a very small amount. Thus, the different cases all represent a specific size regime with no or little overlap to the other cases. Especially, the very small NAT particles (0.5 and 1.0 µm) can be completely distinguished from the particles with other median radii, since all of the detected spectra in this size range fall into case-1. The other two cases have overlap with each other.

The total detection capacity with respect to the simulations is also very good. For median radii up to 6.0 µm nearly all spectra that have been detected as cloud spectra can be detected as spectra influenced by NAT particles. Only for 2.5 and 3.0 µm a few cloud spectra cannot be identified as NAT spectra. These spectra all have a larger CI value ($> 3.0$, optically thinner) and

for larger CI values the separation between NAT and the other PSC types degrades (compare Fig. 3). Thus, a small part of the cloud spectra influenced by NAT cannot be distinguished from STS and ice. When the CI threshold value is reduced to 3.0, the detection itself and also the separation between the different size regimes improves, as shown in Figure 4 b). Now all spectra for median radii up to 6.0 μm are identified as NAT. Additionally the separation between the different sizes is now better and case-3 only detects radii $\geq 3.5$ μm. Independent of the CI threshold value only a part of the simulations for the largest median radius of 8.0 μm are detected as NAT (CI $< 5.0$: $\sim 30\%$; CI $< 3.0$: $\sim 40\%$). This is caused by the decreasing magnitude of the radiance decrease from $811\ \mathrm{cm}^{-1}$ to $826\ \mathrm{cm}^{-1}$ for increasing median radius (see Sect. 3.1). The improvement of the detection and discrimination when using a CI threshold of 3.0 shows that it is not advisable to use a larger threshold when analysing observations. According to the different size ranges of the three cases, the cases are hereafter denoted as small NAT (sNAT: $\leq 1.0$ μm), medium NAT (mNAT: $1.5 - 4.0$ μm), and large NAT (lNAT: $\geq 3.5$ μm). Compared to the former method where only the NAT index-1 is used (e.g. Spang and Remedios, 2003; Höpfner et al., 2006; Spang et al., 2012, 2016, 2018) our new approach with three NAT indices enables an improved detection capacity as more NAT clouds can be detected. For our simulations the improvement is about a factor of 1.78 (approximately 190000 cloud spectra identified as NAT with all new indices and about 108000 spectra identified as NAT when only using index-1) for a CI $< 3.0$.

### 3.2.2 Influence of tropospheric clouds

The influence of tropospheric clouds was intensively studied during the analysis of the radiative transfer simulations for various PSC situations for the MIPAS-Env satellite instrument (Spang et al., 2012, 2016) and the results can be transfered to here. A tropospheric cloud below the PSC mainly influences the absolute radiance values where the cold tropospheric cloud leads to lower overall radiance values. The appearance of the spectral signature (typical peak, shifted peak, or step-like behaviour) in the region $810\ \mathrm{cm}^{-1}$ to $820\ \mathrm{cm}^{-1}$ remains nearly unchanged. Simulations by Woiwode et al. (2019) also showed these effects. As our detection method is based on colour ratios, i.e. relative effects, the change of the absolute radiance values does not significantly influence the analysis. Especially in the case of STS or ice, which show no distinct spectral features in the region $810$–$820\ \mathrm{cm}^{-1}$, the data points in the scatter plots only slightly shift along the correlation lines/regions for the specific PSC type. Thus, the separation lines, which are defined using the STS and ice simulations (see Fig. 3), are still valid in such cases. Consequently, a false detection by an assignment of STS or ice to a NAT class is not possible. For NAT clouds the impact of underlying clouds slightly depends on the median radius of the particles, as the scattering contribution increases with radius (see Fig. 2 c)). For median radii $\leq 1.0$ μm an influence of a tropospheric cloud is negligible, as the spectra are dominated by absorption/emission. For larger median radii a little change of the features can occur and a little change of the positions of the data points in the scatter plots is possible. Possible effects eventually are an assignment of NAT with a median radius of 1.5 μm to the class sNAT (upward shift of data points for the difference NAT index-1 – NAT index-2) or a missing detection for lNAT (downward shift of data points for NAT index-3). Thus, a very little uncertainty of the class boundaries cannot completely be ruled out. However, the majority of all possible situations will still be detected and classified correctly.

### 3.2.3 Mixed NAT/STS clouds

We additionally simulated mixed NAT/STS clouds to evaluate the influence on the spectra and especially the performance of our classification into different size regimes. Here, we only simulated a subset of all possible combinations, as the spectral behaviour for each median radius is very similar independent of e.g. the bottom altitude of the cloud or the thickness (see Sect. 2.4).

If STS is present in addition to NAT, the relative magnitudes of the spectral features caused by NAT particles get smaller and therefore the separation between these mixed clouds and STS is also restricted compared to a pure NAT cloud. Figure 5 a) shows example spectra for PSCs of pure NAT (solid lines in red for 0.5 μm and blue for 1.0 μm) and PSCs with the same NAT and additionally STS (dashed lines) to illustrate this effect. In a scatter plot of the NAT indices versus the CI (compare Fig. 3 a) and b)) the simulation results for the NAT/STS mixtures are closer to the separation line or even below compared to pure NAT. When the NAT detection and classification procedure (described in the previous subsection) is applied to the simulation results for the mixed clouds, the good discrimination between the small and medium size particles remains. This discrimination relies on the difference of NAT index-1 and index-2, where both indices are influenced in the same way. Therefore, the sign of the difference remains the same because this depends on the position of the spectral peak, which is not affected by the additional STS (see Fig. 5 a)). For a CI value below 3.0 about 97.5% of the simulated cloud spectra with median radii 0.5 and 1.0 μm are correctly detected as sNAT and only about 0.3% are wrongly detected as mNAT (the remaining 2.2% are not detected as NAT). In some cases STS completely masks the spectral features caused by NAT and NAT is not detectable any more. However, for a CI value below 3.0 more than 90% of the cloud spectra can still be identified as containing NAT in the entire simulated size range. The proportion typically decreases with increasing median radius, i.e. more NAT influenced spectra at 3.0 μm are missed than at 0.5 μm, and also will decrease with increasing STS volume density or decreasing $HNO_3$ VMR inside the PSC. In a nutshell, in mixed STS-NAT-clouds fewer scenarios are identified as containing NAT, because the STS reduces the amplitude of the characteristic NAT signature. But, if a scenario of our simulations was classified as NAT, the size attribution remains as reliable as in the pure NAT scenarios.

### 3.2.4 Bimodal NAT clouds

In addition, we also simulated PSCs using bimodal NAT particle distributions with a main focus on the small and medium size particles and the separation between those. Here, also a subset of all possible combinations was simulated (see Sect. 2.4).
The spectra simulated for the bimodal NAT particle size distributions are typically some kind of mixture of the spectra for the corresponding unimodal distributions, thereby the $HNO_3$ VMR ratio plays an important role. Figure 5 b) shows an example for a spectrum simulated for a bimodal NAT particle distribution (black line). The $HNO_3$ VMRs were 5 ppbv in each mode with median radii of 2.5 μm and 6.0 μm. The two corresponding spectra for the unimodal size distributions are shown in red and blue, respectively. Obviously, the spectrum for the bimodal size distribution is a mixture of the other two. When the $HNO_3$ ratio for both modes is changed to 70/30 or 30/70 the spectrum looks more like the spectrum for that unimodal size distribution that dominates the bimodal distribution, because of more $HNO_3$ in the corresponding size range.

We applied our classification procedure also to these simulations for bimodal size distributions with the following results. When the first mode dominates (70% HNO$_3$), nearly all cloud spectra simulated for a median radius of 0.5 or 1.0 µm are still

detected as sNAT (about 99% sNAT and 1% mNAT), and thus classified correctly. All spectra that are identified as cloud (CI < 3.0) are also identified as NAT containing PSCs. For a HNO$_3$ ratio of 50/50 or 30/70 the influence of the second mode increases such that more and more spectra are detected as mNAT and a small part as lNAT. In the case of the ratio 50/50 about 50.5% are classified as sNAT and 49.5% as mNAT and for a ratio of 30/70 21% are classified as sNAT, 77% as mNAT, and 2% as lNAT. When combined with larger NAT particles in the second mode ($\geq$ 4 µm) a part of the cloud spectra (CI < 3.0) are not detected

as NAT because neither a spectral peak nor a step-like behaviour can be detected (50/50: 0.5% and 30/70: 10%). However, these cases are only a few percent of all cloud spectra in our simulations. When the first mode has a median radius between 1.5 – 2.5 µm about 99% of the simulated spectra are identified as mNAT (1% lNAT, when the median radius in the second mode is 5.0 or 6.0 µm) independent of the ratio of the HNO$_3$ VMRs. Furthermore, only a few per thousand of the spectra identified as clouds cannot be classified as NAT containing PSCs. In total, our new classification scheme delivers very reasonable results

even in the case of bimodal NAT particle size distributions.

### 3.3 Detection of ice

The detection of ice uses the fact that in the presence of ice a large radiance decrease from about 833 cm$^{-1}$ to 949 cm$^{-1}$ can be observed (compare Fig. 1). This clear decrease is only observed in the case of ice or in the presence of very small NAT particles. Spang et al. (2012, 2016) used this spectral behaviour to detect ice clouds in satellite measurements of MIPAS-Envisat. The

authors used the brightness temperature (BT) difference between the two spectral regions. Here, we adopt the method for the CRISTA-NF observations. Because of the different viewing geometry, spectral resolution and the different definition of the CI for the two instruments, the separation lines have to be newly defined. The spectral regions used for the BT difference are 832.0 – 834.0 cm$^{-1}$ (MW2) and 947.5 – 950.5 cm$^{-1}$ (MW7). In a scatter plot of the BT difference against the CI, the simulated ice spectra clearly separate from other particle types (Fig. 6). Obviously, spectra that are influenced by ice clearly separate from

STS and nearly all NAT particles when the cloud is optically thick enough (low CI). For larger CI values the separation gets smaller. The only particle type that can produce similar values of the BT difference are very small NAT particles with median radii of 0.5 µm (yellow colours in Fig.6 a)), but these particles can be safely filtered out with the method described before in Sect. 3.2. Consequently, the BT difference is a very robust method to detect ice particles in PSCs.

### 3.4 Detection of STS

The spectra for STS show neither a local spectral feature like the spectra for NAT nor a broadband spectral feature such as the spectra for ice. Thus, the detection of STS can not be achieved by using a unique spectral behaviour. In practice the detection procedure is as follows. Firstly, the NAT detection methods are used to detect observations of NAT particles and to distinguish between the three size regimes. Secondly, the BT difference method is applied to the observations to detect ice. Finally, the observations inside PSCs that are neither detected as NAT nor as ice are categorised as STS. It is not necessarily the case that the spectra categorised as STS are solely influenced by STS. It is possible that NAT or ice also were present in the PSC but the

additional amount of STS was enough to minimise the spectral features such that NAT or ice is not definitely detectable any more. Furthermore, the small amount of very large NAT particles that cannot be distinguished from STS and ice will also fall in the category STS.

## 3.5 Bottom altitude of the PSCs

Further important quantities with respect to cloud detection are the vertical thickness and the position of the cloud defined by the top and bottom altitude. Here, we present a method to determine the bottom altitude of the observed cloud. The cloud index, which is used to detect optically thick conditions caused by clouds (or aerosol), shows characteristic vertical changes in the presence of clouds. These changes are used for the detection of the cloud bottom altitude.

The left panel of Fig. 7 shows examples of CI altitude profiles for clouds with different vertical thicknesses. When the complete
cloud is located below the flight altitude (yellow colour) the CI largely drops to low values when entering the cloud from above (or slightly above because of the FOV). In the case the flight altitude is inside the cloud the CI is already low at flight altitude. Then the CI typically further decreases with decreasing altitude inside the cloud and the minimum CI value is reached close to the bottom altitude. Only in some cases when the vertical thickness is larger (4 km or 8 km) the CI minimum can be located somewhere inside the cloud. When leaving the cloud the CI increases again. These vertical changes of the CI can be best
illustrated with the gradient of the CI (right panel in Fig. 7). Slightly above the cloud top the vertical gradient maximises and slightly below the cloud bottom the gradient reaches its minimum value. In contrast to the CI minimum, where larger deviations to the cloud bottom can occur, the minimum of the gradient is always slightly below the real bottom altitude of the cloud.

Figure 8 summarises the results for the simulated clouds with a bottom altitude below flight altitude. In the case of the CI minima (full circles) there are sometimes larger deviations from the real bottom altitude when the cloud is vertically thick (red
and light blue full circles). The CI gradient minima (full diamonds) are always located close to the real bottom altitude, at the first or the second measurement below. The possibility to observe the gradient minimum at the second measurement below the cloud increases with decreasing altitude of the cloud, because the vertical extent in metres of the FOV is larger at lower altitudes. In our simulations this occurs only at bottom altitudes of 13.0 and 14.0 km. In summary, the CI minimum and the gradient minimum enclose the real bottom altitude. A large gap between the two minimum values indicates the observation of
a cloud with a larger vertical thickness.

There are further restrictions for the determination of the bottom altitude. In some cases the observations run into saturation and the CI values below the cloud bottom altitude converge at a low value. Additionally, in some other situations the CI values below the bottom altitude only show a linear increase and not the larger change directly below the bottom altitude. In both cases the CI minimum and the gradient minimum cannot sufficiently be determined and used for the detection of the bottom altitude
of the cloud. An effective way to sort out possibly affected observations is the use of a threshold value. A CI minimum below 1.25 we derived from the simulations. Such low values of CI only occur for a part of the ice clouds simulated here (typically for largest volume densities of 50 and 100 $\mu m^3/cm^3$; with increasing vertical thickness also some simulations with 5 and 10 $\mu m^3/cm^3$ were affected) and for a few of the NAT and STS clouds with large $HNO_3$ VMRs ($\geq 11$ ppbv) or volume densities ($\geq 5$ $\mu m^3/cm^3$) combined with a large vertical thickness ($\geq 4$ km). Secondly, when the cloud is optically and vertically thin,

the location of the cloud bottom is not detectable as well. In order to select only clouds that are optically thick enough for detection we used only simulation spectra with CI < 5.0.

Furthermore, more special situations can be considered, PSC only above flight altitude and layered PSCs. When the PSC is located above the flight altitude the CI typically increases with decreasing altitude from flight altitude downwards, because the length of the line of sight inside the PSC decreases also. Thus, there is not the typical decrease of the CI to a local minimum,

but rather a continuous increase from a rather low CI value compared to clear air at flight altitude. Both, the CI and the CI gradient, typically show the lowest values at the points closest to the flight altitude. In the case of layered PSCs, where PSCs of different type are lying above each other, the results depend on the optical thickness of the different layers. On the one hand, the CI gradient minimum is still located at the bottom of the complete PSC, when the lower layer is optically thick enough and the transition to the cloud free atmosphere shows the largest increase in CI. On the other hand, when the lower layer is optically

very thin, it can occur that the CI gradient minimum is located directly below the upper layer. In this case the transition from the upper layer to the lower layer leads to a larger increase in CI than the transition from the lower layer to the cloud free atmosphere.

## 4 Application to the CRISTA-NF measurements

This section shows the application of the methods derived from the simulations to observations by the CRISTA-NF instrument

for all flights during RECONCILE, where PSCs were observed. These PSC observations were carried out during the first five flights between 17.01.2010 to 25.01.2010. For one selected flight (local flight 3: 22.01.2010) we additionally show more details of the results to demonstrate the capability of the instrument and the methods.

### 4.1 Analysis of the CRISTA-NF measurements

We applied the dection methods for NAT and ice using the NAT indices and the BTD to the measurements by CRISTA-NF for

flight 1 – 5. The results for the different scatter plots are shown in Fig. 9 a) – e). Note here, for flight 2 – 5 the measurements were filtered so that only measurements inside PSCs have been taken into account (more details see Sect. 4.2). In the case of flight 1 this filtering was not possible as many profiles show very low CI values (see Sect. 3.5) and, thus, simply all measurements between flight altitude and 14 km are shown. The observations separate into two different situations. First, during flight 1 a clear signal of ice was detected, which can been seen in Fig. 9 e), and no indication of NAT. For all NAT indices vs. CI the

data points stay below the separation lines, which were derived from the simulation results, in the same way as the simulation results for ice (see Fig. 3). This shows the good correspondence between simulations and observations and gives additional confidence in the simulation results and the derived separation lines using these results. The ice observation is supported by ECMWF data (not shown), which show temperatures low enough for ice formation above the flight altitude at about 20 km and above, and CALIPSO observations that show ice PSCs above 20 km a few hours before the flight (personal communication:

480 M. Pitts). Thus, CRISTA-NF observed the ice PSC from below. As ice PSCs are typically optical thicker than other PSCs, this explains the lower values of CI observed during this flight. Second, the flights 2 – 5 all show signatures of NAT (see Fig. 9 a)

and b)). As the differences between NAT index-1 and index-2 are below the separation line (see Fig. 9 c)), these observations are categorised as mNAT (for flight 5 a few spectra are categorised as lNAT). The difference of the spectra for these two PSC types is additionally illustrated in Fig. 9 f) with example spectra from flight 1 (blue colours) and flight 3 (red colours). They are scaled for better comparability. Obviously, the spectra in flight 3 show a shifted NAT peak and the spectra in flight 1 a large negative gradient towards larger wavenumbers, which in the absence of the of the $820 \ \mathrm{cm}^{-1}$ NAT feature clearly shows the influence by ice (compare Fig. 1).

Because of the strong radiance enhancement caused by the clouds, the dominating uncertainty is the relative uncertainty (noise uncertainty) of the measurements, whereas a systematic uncertainty (e.g. an offset) plays a minor role. The relative uncertainty for the radiances typically observed during atmospheric measurements is about 1–2% for CRISTA-NF (Schroeder et al., 2009). Because of the calculation of ratios, the relative error can add up, but is still only a few percent. In the worst case the maximum uncertainty is about 2-4%. For the analysis of the PSC observations this has a different effect depending on the CI and the NAT-index. For smaller values of the CI and NAT indices the resulting absolute uncertainties of these quantities determined from the relative uncertanties are smaller than for larger values. Thus, the methods work better for smaller CI values. This suggests to use a threshold value for the CI and to only analyse observations with a CI below this threshold as it was done in Sect. 3.2. Furthermore, the same spectral windows are used for CI and NAT index-1/2. In this case some of the uncertainty will cancel, as the data points will shift similar to the correlation regions/lines, i.e. a smaller CI (because of a smaller randiance in the $CO_2$-window) will be accompanied by a larger NAT index and vice versa. However, many observations, especially for flight 3 and 5, show deviations from the separation line larger than a few percent and a clear separation between ice and NAT (see Fig. 9 a), b)). Furthermore, they show a compact correlation and not a spread as it would be expected for large noise errors.

## 4.2 Detailed Results for flight 3

We show more detailed results for the detection of the bottom altitude and the detection and discrimination of PSCs exemplarily for the RECONCILE local flight 3, which took place on 22.01.2010. The flight started in Kiruna (Sweden) and the whole flight was located northward of Kiruna inside the polar vortex.

### 4.2.1 Bottom altitude

During the RECONCILE local flight 3 the aircraft flew through PSCs. Thus, the CRISTA-NF instrument made measurements inside PSCs during a large part of the flight. The behaviour of the CI and the CI gradient can now be used to detect the bottom of the PSCs. Figure 10 shows the CI and the CI gradient for two selected altitude profiles that have been measured inside PSCs. These two profiles show the behaviour as expected from the simulations. In case of profile 102 (right panel in Fig. 10) the CI becomes smaller from flight altitude downwards as long as the measurements are inside the cloud (compare red and yellow profile of CI in Fig. 7). The CI gradient shows the largest negative value one sampling step below the CI minimum. According to the simulation results the CI gradient minimum is located below the cloud whereas the CI minimum is located inside the cloud. Thus, in this example an accurate detection of the bottom altitude is possible (profile 102: $\sim$17.3 – 17.4 km). In the case of profile 80 the behaviour of CI and CI gradient is very similar. Only the altitude difference between the CI minimum and the

CI gradient minimum is larger than only one sampling step (profile 80: ~17.2 – 17.5 km). This can be caused by two effects. Firstly, the PSC is inhomogeneous and this causes the increase of the CI above the CI gradient minimum, which is supposed to be located directly below the PSC. Secondly, in the case of vertically largely extended clouds this behaviour is also observed in the simulations (compare Fig.7 blue curves). If the vertical extent is responsible for the difference between the two minima, this would suggest that the vertical extent is larger than 2 km. However, the difference between the two minima is only about 300 m. Thus, the detection of the bottom altitude, which is located between these two points, is still very accurate.

Figure 11 a) shows the cross section of the CI for the complete flight. During the middle section of the flight the aircraft crossed PSCs as can be seen by the low CI values (blue colours) at and directly below the flight altitude. Beneath this region the CI values are larger again, which indicates cloud free conditions below the PSCs. The green and the magenta line in Fig. 11 a) mark the CI minima and the CI gradient minima, respectively. Only profiles where both minima could sufficiently be determined are considered. Similar to the results for the two selected profiles (compare Fig. 10) the two minima are primarily located in the region between 17 and 17.5 km. Thus, for most profiles a good estimate of the bottom altitude of the cloud is achieved.

### 4.2.2 PSC classification

During a large part of the flight numerous spectra below the bottom altitude of the cloud (see green and magenta lines in Fig. 11 a)) would be detected as PSC influenced spectra. This is expected as a large part of the line of sight (LOS) is still inside the PSC and, thus, the spectral features caused by the PSC particles are visible in the spectra. Since the cloud bottom height was determined by the CI gradient method, we restricted the analysis to the altitude range between flight altitude and the CI gradient minima. Figure 11 b) shows the cross section of the detected PSC types. Additionally, only spectra with a CI value below 3.0 are considered (Fig. 11 b)).

During the complete flight only medium sized NAT and STS were observed (orange and light blue colours in Fig. 11 b)). Figure 12 shows example spectra inside the PSCs at about 12:10 UTC for illustration, which show a shifted NAT feature at about 816 $cm^{-1}$, and thus a classification as mNAT is expected. The new method reliably detects spectra showing such a shifted NAT feature. Most of the observations detected as STS are located in the second half of the PSC observation from the flight altitude downwards. This is also in accordance with the spectra that have been measured in this region. These spectra (see Fig.12) show a clear shifted NAT feature only at altitudes a few hundred metres below flight altitude. Directly below the flight altitude the NAT feature can hardly be seen. This does not necessarily mean that no NAT was present in this altitude region, but the STS contribution to the measured radiances was that large, thus all other particle type signatures were masked out. Additionally, the CI values in this region are lower directly below flight altitude (compare Fig. 11 a)) compared to the first part of the PSC. In addition to the smaller distance between the CI minima and the gradient minima in the second part of the PSC this suggests that the vertical extent of the cloud is smaller compared to the first part or that the PSC is optically thinner.

In summary the methods derived in Sect. 3 are able to give a complete picture of the observed PSC. The PSC and the bottom altitude of the cloud are clearly detected and the new and improved detection method enables the classification of medium size NAT particles and STS during RECONCILE flight 3.

## 5 Discussion

Small NAT particles cause a distinct spectral peak in infrared limb emission spectra at about 820 cm$^{-1}$. This peak has already been observed in satellite measurements since the 1990's (Spang and Remedios, 2003; Höpfner et al., 2006). We showed with our simulations that the appearance of the NAT feature changes with changing particle size. The spectral peak (small NAT) transforms to a shifted peak (medium NAT) and, finally, to a step-like behaviour of the spectrum (large NAT) with increasing median radius of the particle size distribution. This change is related to the different proportions to which scattering and absorption/emission contribute to the total radiance (see also Woiwode et al., 2016).

The change in the appearance of the feature can be used to distinguish between different size regimes. Our new approach enables the differentation between small NAT (0.5 – 1.0 μm), medium NAT (1.5 – 4.0 μm), and large NAT ($\geq$ 3.5 μm). As the complete analysis method is based on color ratios, i.e. relative differences, the method searches for spectra that qualitatively show same features as in the simulations. Possible differences concerning the absolute radiance values (e.g. influenced by vertical thickness, optical thickness of the PSC etc.) only lead to changes inside the correlation region for a specific particle type or size. In contrast to the former method where only one NAT index is used (detection of NAT < 3.0 μm) (see Spang and Remedios, 2003; Höpfner et al., 2006; Spang et al., 2016) this improved method leads to a larger detection capacity as more NAT containing clouds can be detected. A part of the medium sized particles (those when only NAT index-2 is above the separation line) and the complete size range of large NAT particles are not detected with the former method. Probably this improvement will diminish the discrepancy between NAT cloud observations by MIPAS-Envisat and the CALIOP (Cloud-Aerosol Lidar with Orthogonal Polarization) instrument onboard CALIPSO (Cloud-Aerosol Lidar and Infrared Pathfinder Satellite Observation; Pitts et al., 2018) in the Northern hemisphere, where much more NAT was observed by CALIOP and the agreement for the NAT observations by both instruments is only 18% (the agreement for NAT observations in the Southern hemisphere is much better with about 73%) (Spang et al., 2018; Tritscher et al., 2020). Consequently, this would conclude a larger probability for the formation of NAT clouds with medium to large radii for the Northern hemisphere compared to Southern hemisphere conditions, which highlights a link to different formation mechanisms and meteorological conditions fostering the formation of large or small NAT particles.

The derived methods can easily be adopted to analyse other observations like that of MIPAS-Envisat, as the spectral behaviour in general is the same. The separation lines would need to be adjusted, because different instruments have different spectral properties and viewing geometries. Thus, the new NAT detection method can be transfered to MIPAS-Envisat, but a set of simulations is necessary to do this. In the case of the ice detection we sucessfully transferred the method used for MIPAS-Envisat (e.g. Spang et al., 2016) to the airborne geometry of CRISTA-NF and refined the separation lines.

In recent publications by Woiwode et al. (2016, 2019) a special step-like behaviour of infrared spectra in the presence of NAT particles was intensively analysed and the authors simulated the observed spectra by the presence of large aspherical NAT particles. Woiwode et al. (2019) showed the large shoulder-like signature at about 820 cm$^{-1}$ and flat radiance behaviour afterwards, which they called a hockey-stick signature, for an example observed by MIPAS-Envisat. The authors assigned this behaviour to be characteristic for large aspherical NAT particles and developed a detection method for this type of spectrum.

The main criteria used for the detection are a difference in the integrated radiance between the window $817.5 – 818.5 \, \text{cm}^{-1}$ and $833.0 – 834.0 \, \text{cm}^{-1}$ above a certain value (to detect the step or shoulder) and a difference in the integrated radiance between the window $833.0 – 834.0 \, \text{cm}^{-1}$ and $960.0 – 961.0 \, \text{cm}^{-1}$ below a certain value (to detect the flat behaviour at larger

wavenumbers). Our simulations agree at the point that large spherical NAT particles (with unimodal distribution) typically show a step-like behaviour but a decrease in the radiance towards larger wavenumbers (compare Fig. 1 reddish colours and Fig. 5 b) blue line), and therefore do not show this hockey-stick signature. But some of our simulations for the bimodal NAT particle size distributions show this typical behaviour. The spectrum in Fig. 5 for the bimodal NAT particle size distribution (black line) exhibits no difference in radiance from about 833 to $960 \, \text{cm}^{-1}$. Furthermore, the step or shoulder is more pronounced compared

to the spectrum for the unimodal size distribution (blue line) and slighty shifted towards larger wavenumbers. Therefore, the radiance decrease from 818 to $833 \, \text{cm}^{-1}$ increases. In our opinion, this spectrum will probably fullfil all criteria used for the detection of large aspherical NAT particles as described in Woiwode et al. (2019). This suggest that large aspherical NAT particles are not necessarily the only possibility to observe such a kind of spectrum or spectral signature that fullfil the criteria of the detection method. Thus, the spectrum alone is possibly not enough to definitely detect aspherical NAT particles. But a

complete picture of the situation involving infrared emission spectra, FSSP (Forward-Scattering Spectrometer Probes; Molleker et al., 2014) observations of the particle size distribution, and information on the $HNO_3$ budget together will presumably be sufficent to make a clear decision. For single spectra this has been done by Woiwode et al. (2016, 2019) and in these special cases aspherical NAT particles led to the best results.

As the differences used in the detection scheme by Woiwode et al. (2019) rely on absolute differences and as CRISTA-NF

and MIPAS-Envisat have different viewing geometries etc., the absolute radiance values and therefore the differences are not comparable. Thus, the detection method cannot be applied to CRISTA-NF observations or simulations. In order to analyse the detection scheme by Woiwode et al. (2019) in more detail simulations for the MIPAS-Envisat instrument are necessary, which is beyond the scope of this paper. However, our simulation results demonstrate that spherical particles can qualitatively produce all the different appearances of the NAT feature (typical feature, shifted feature, step-like behaviour) in the considered spectral

range including those previously attributed to highly aspherical NAT. We think that this is an important point to consider as it implies that IR limb emission measurements alone are not sufficient to prove the existence of aspherical NAT, in particular for cases without additional information on the atmospheric state and cloud particle size distribution. An approach using additional information about the particle size distribution and the $HNO_3$ budget and a quantitative comparison as done by Woiwode et al. (2016, 2019) will help to gain more information about particle shape. But such an approach is only feasible for local case

studies and not for global analyses. Our approach of grouping the NAT particles into 3 size bins based on the spectral signature we do not consider affected by the particle shape for the following reasons. The small particles ($< 3 \, \mu$m) are considered not or only slightly aspherical. For our approach we used the term 'small' for particle size distributions with median radii $\leq 1.0 \, \mu$m. Also Woiwode et al. (2019) use the method by Höpfner et al. (2006), which is based on simulations using spherical particles, to detect these small particles. Further, a substantial fraction of our 'medium' size group (median radii $\leq 4 \, \mu$m) can be considered

nearly aspherical. The shifted NAT signature reported by Woiwode et al. (2016) for highly ashperical NAT is bi-modal with median radii of 2 and $4.8 \, \mu$m, where the latter is dominating the signal. This means that the shifted NAT signature that was

simulated by using this bi-modal size distribution in combination with aspherical particles, has only been observed in the presence of larger NAT particles. Further, Woiwode et al. (2019) (Fig. 1) show that for median radii $\geq 5.0\ \mu$m a step-like NAT-feature can be observed. Hence, our result that the shift of the NAT signature is associated with NAT particle size is in line with the results of Woiwode et al. (2016, 2019). Due to particle asphericity our size bin boundary between 'medium' and 'large' has uncertainties and the numbers given here are valid under the assumption of spherical particles.

Grooß et al. (2014) presented a measured size distribution for a part of flight 3. The maximum in the distribution lies at a radius of about $2.5 - 3\ \mu$m and a second smaller maximum at about $5 - 6\ \mu$m is visible. Our results for this flight show mNAT particles ($1.5 - 4\ \mu$m). This is in a reasonable agreement considering the limits of the comparisons. Note here, that the particle size distribution is observed solely at flight altitude and CRISTA-NF observes much larger air masses. Additionally, it is likely that the larger particles are more located at lower altitudes and the number density of these large particles will decrease with increasing altitude due to sedimentation. CRISTA-NF is also influenced by the air masses above flight altitude. Molleker et al. (2014) presented particle size distributions for the flights 4 and 5. The maximum for flight 4 is at about $2.5 - 3\ \mu$m and additionally also larger particles were detected. Our results for this flight show mNAT, which is in agreement under the aforementioned restrictions for the comparison. For flight 5 different size distributions are presented, which show two maxima, one at about $2.5 - 3\ \mu$m and a second at about $5 - 6\ \mu$m. This flight shows the largest amount of large particles. Our results for flight 5 show mainly mNAT and a few lNAT. This is again in agreement within the scope to the limits of the comparison. However, a detailed comparision would need additional information e.g. on the vertical change of the size distribution and the vertical extent of the cloud.

## 6   Summary and conclusions

We performed a large set of radiative transfer simulations of infrared limb emission spectra in the presence of polar stratospheric clouds of different types (NAT, STS, ice). All simulations have been performed for the viewing geometry and spectral properties of the CRISTA-NF instrument. These simulations build a new data base that is used for the analysis of PSC spectra to develop and refine detection and discrimination methods.

We showed with our simulations that the NAT feature changes from a spectral peak at $820\ \mathrm{cm}^{-1}$ (small NAT) to a shifted peak (medium NAT) and, finally, to a step-like behaviour of the spectrum (large NAT) with increasing median radius. Based on this behaviour we developed an improved method to detect NAT particles, which for the first time allows the discrimination of three different size regimes: small NAT ($0.5 - 1.0\ \mu$m), medium NAT ($1.5 - 4.0\ \mu$m), and large NAT ($\geq 3.5\ \mu$m) under the assumption of spherical particles. By using color ratios, i.e. relative differences, the method qualitatively compares observed spectra with the complete simulated data base. This new detection method will also improve the analysis of other observations by infrared limb emission sounder such as MIPAS-Envisat. The ice detection method was adopted from former studies (MIPAS-Envisat) and the separation lines were newly defined. Additionally, we developed a new method to detect the bottom altitude of the clouds. This method uses the gradient of the CI, which minimises shortly below the real bottom altitude. As the minimum of the CI itself is located inside the cloud (typically close to the bottom), these two quantities give a good estimate for the bottom

altitude. This method can surely be transfered to other cloud observations such as cirrus clouds and aerosol layers and to other airborne instruments measuring in the same wavelength region like e.g. GLORIA (Gimballed Limb Observer for Radiance Imaging of the Atmosphere; Riese et al., 2014). A prerequisite for a sucessful usage of the method is a small FOV like that of CRISTA-NF. Larger FOVs will lead to a larger vertical averaging of the measurements and thus to a reduction of the detection capabilities. This can already be seen for the CRISTA-NF instrument, where at lower altitudes (14 km and below) the minimum of the CI gradient can move further away from the real bottom altitude.

Finally, we applied the new methods to observations of CRISTA-NF during the RECONCILE local flight 3. The results show a polar stratospheric cloud that has been crossed during the flight by the aircraft and extends downward to about $17 - 17.5$ km. The PSC contained NAT particles, which could be classified to be of medium size ($1.5 - 4$ μm) as in the spectra always a shifted NAT feature has been observed. This shifted feature is detected by the new method. Moreover, using the method developed here a new data set of PSC observations and classification can be obtained. This new data set will help to improve the results of trace gas retrievals in the presence of PSCs by integrating realistic PSC extinction spectra into the retrieval process. Furthermore, the gained data will help to improve the representation of PSCs in model simulations.

*Code and data availability.* The JURASSIC code is available at https://github.com/slcs-jsc/jurassic-scatter. The simulation results are available at https://datapub.fz-juelich.de/slcs/cloud-spectra/psc-crista-nf/ (Kalicinsky et al., 2020). The CRISTA-NF level 1b data can be obtained by request to the corresponding author.

*Author contributions.* The setup of the simulations (background atmosphere, cloud scenarios, etc.) was compiled and discussed with all authors CK, SG, and RS. The simulations have been mainly performed by SG and the analysis was mainly done by CK under intensive discussions with all authors. CK wrote the manuscript with contributions of the two other authors SG and RS.

*Competing interests.* The authors declare that they have no competing interests.

*Acknowledgements.* The work by Christoph Kalicinsky was funded by the German Science Foundation (DFG) under the grant number 4118/2-1. The authors gratefully acknowledge the Gauss Centre for Supercomputing e.V. (www.gauss-centre.eu) for funding this project by providing computing time through the John von Neumann Institute for Computing (NIC) on the GCS Supercomputer JUWELS at Jülich Supercomputing Centre (JSC). We thank the Canadian Space Agency for access to the ACE-FTS data. We gratefully acknowledge the European Centre for Medium Range Weather Forecast (ECMWF) for proving the ERA-Interim data. We thank C.M. Volk and the HAGAR team for the access to the HAGAR data. We additionally thank M. Diallo for providing the $CO_2$ data product.

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

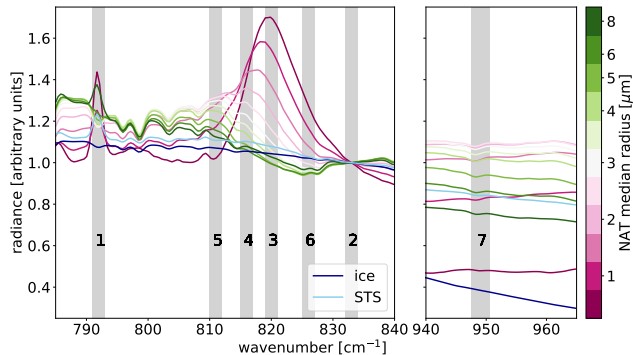

**Figure 1.** Selected simulation results for infrared spectra in the presence of polar stratospheric clouds consisting of NAT particles with different particle median radii, STS, and ice. The spectra have been scaled using the mean radiance in the spectral window $832.0 - 834.0$ cm$^{-1}$ (MW2) such that the radiance for all spectra equals 1 in this window. The gray vertical bars mark the micro windows (MW) used during the analysis. They are numbered from MW1 to MW7.

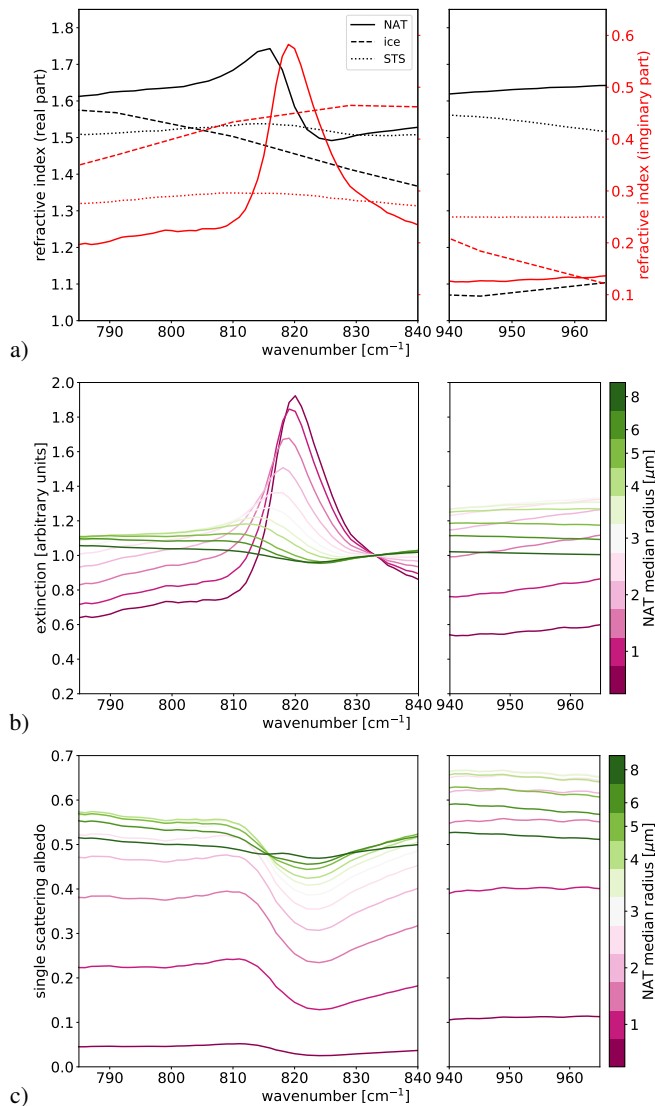

**Figure 2.** a) Refractive indices for NAT (solid lines), STS (dotted lines), and ice (dashed lines). The real parts are shown in black and the imaginary parts in red with a second axis to the right. The refractive indices for NAT, STS, and ice were taken from Biermann (1998) with refinement in Höpfner et al. (2006), Biermann et al. (2000), and Toon et al. (1994), respectively. b) Extinction spectra for NAT with different median radii. The spectra have been scaled using the mean value in the spectral window $832.0 – 834.0 \ \mathrm{cm}^{-1}$ such that the extinction for all spectra equals 1 in this window. c) Single scattering albedo for NAT with different median radii.

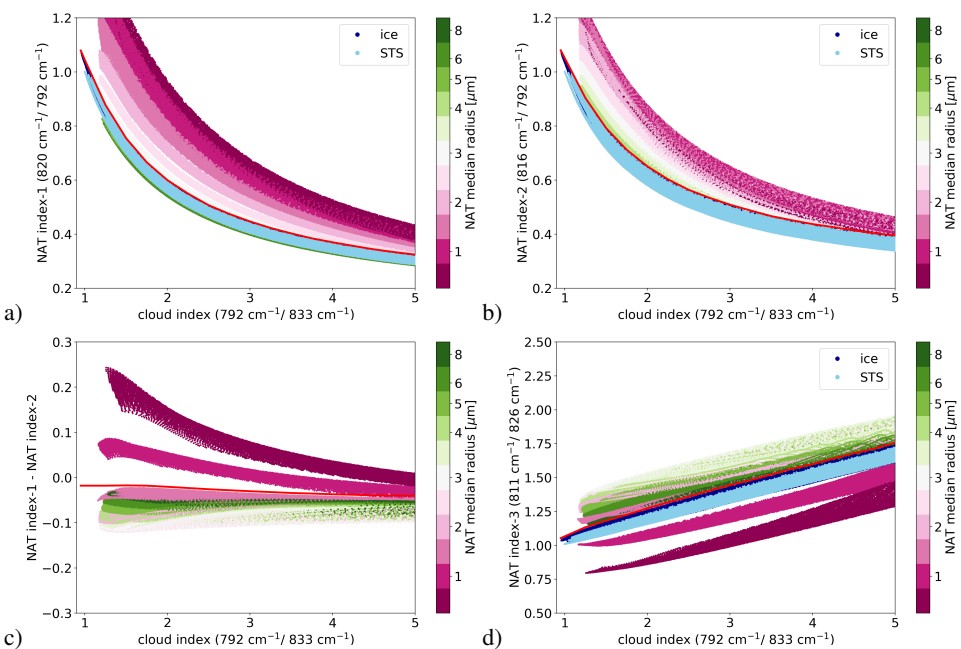

**Figure 3.** Correlations between different NAT indices and cloud index for the simulated PSC scenarios. a) NAT index-1 $(819 - 821$ $\mathrm{cm}^{-1})/(791 - 793 \ \mathrm{cm}^{-1})$; b) NAT index-2 $(815 - 817 \ \mathrm{cm}^{-1})/(791 - 793 \ \mathrm{cm}^{-1})$; c) NAT index-1 − NAT index-2; d) NAT index-3 $(810$ $- 812 \ \mathrm{cm}^{-1})/(825 - 827 \ \mathrm{cm}^{-1})$. The red lines show the separation lines, which mark the upper envelope of the regions of STS and ice (in a), b), and d)) or the region of medium and large NAT (in c)). Simulation results for ice and STS are in shown in dark and light blue, respectively.

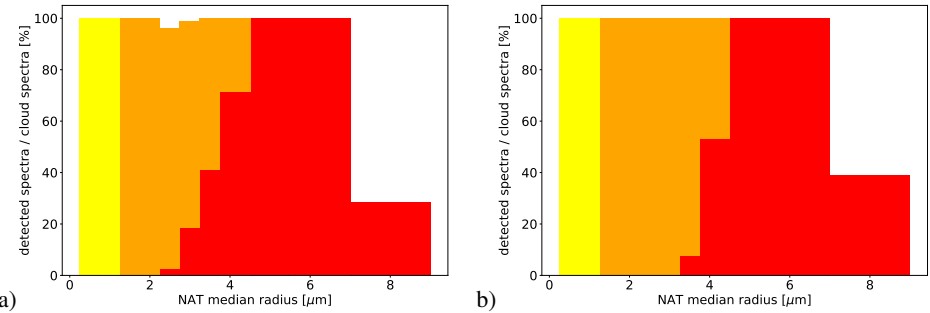

**Figure 4.** The histograms of the proportion of the detected spectra (NAT) per simulated cloud spectra in each size bin. The colours illustrate the different cases 1 − 3. Case-1 (sNAT) is shown in yellow, case-2 (mNAT) in orange, case-3 (lNAT) in red. For the description of the different cases see details in text. In panel a) the CI threshold value to detect a spectrum as cloud spectrum is 5.0 (in total about 365000 cloud spectra) and in panel b) 3.0 (in total about 197000 cloud spectra).

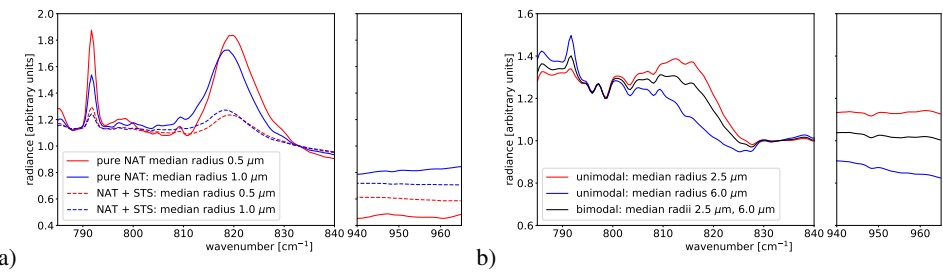

**Figure 5.** Selected spectra for NAT/STS mixed clouds (a)) and bimodal NAT particle size distributions (b)). The spectra were scaled using the mean radiance in the spectral window $832.0 - 834.0$ cm$^{-1}$ such that the radiance for all spectra equals 1 in this window. a) The solid lines show spectra for unimodal NAT particle size distributions. Red: median radius 0.5 µm and 10 ppbv HNO$_3$; Blue: median radius 1.0 µm and 10 ppbv HNO$_3$. The dashed lines show the NAT/STS mixed clouds. The amount of NAT is the same as for the pure NAT simulations and the volume density of STS is 10 µm$^3$/cm$^3$ in both cases. b) The red and blue lines show the spectra for unimodal size distributions with median radius 2.5 µm and 6.0 µm, respectivley. The amount of HNO$_3$ is 10 ppbv in each case. The black line shows the simulation results for a bimodal size distribution with the median radii 2.5 µm and 6.0 µm and 5 ppbv HNO$_3$ in each mode.

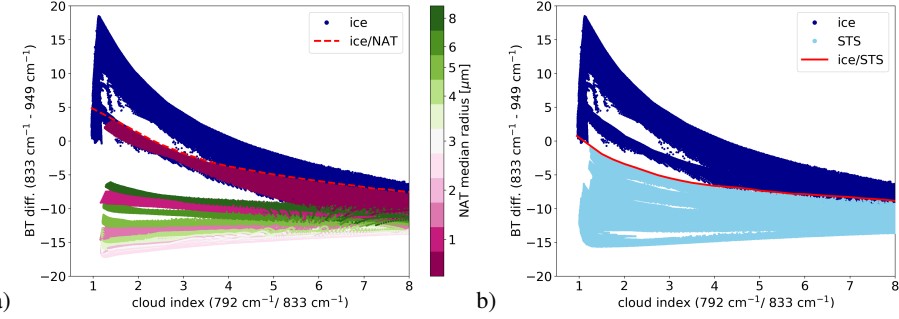

**Figure 6.** Correlation between the BT difference $(832 - 834$ cm$^{-1}) - (947.5 - 950.5$ cm$^{-1})$ and cloud index. The red solid line shows the separation line between ice and STS and the dashed line marks the separation between ice and NAT.

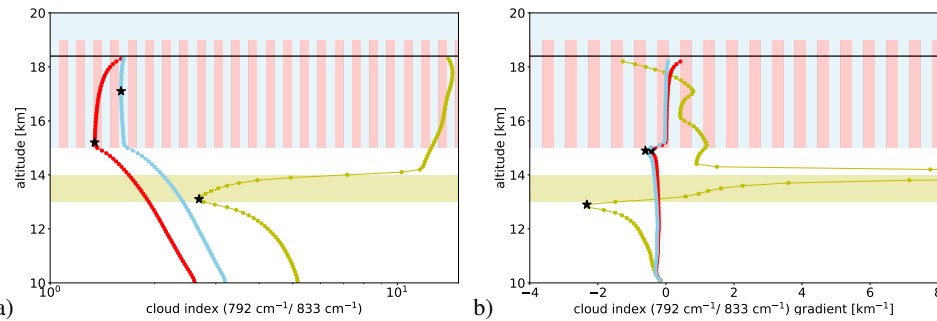

**Figure 7.** Vertical profiles of the cloud index (left) and the vertical gradient of the cloud index (right) for clouds with different vertical thicknesses. The colour coding shows the vertical thickness (yellow: 1 km, red: 4 km, light blue: 8 km up to 23 km). The $HNO_3$ VMRs inside the NAT PSCs are 5 ppbv for the 1 and 8 km thick clouds and 10 ppbv for the 4 km thick cloud. The corresponding shaded areas illustrate the vertical extent of the clouds. The black stars mark the altitudes of the CI minima and the CI gradient minima. The black horizontal line shows the flight altitude.

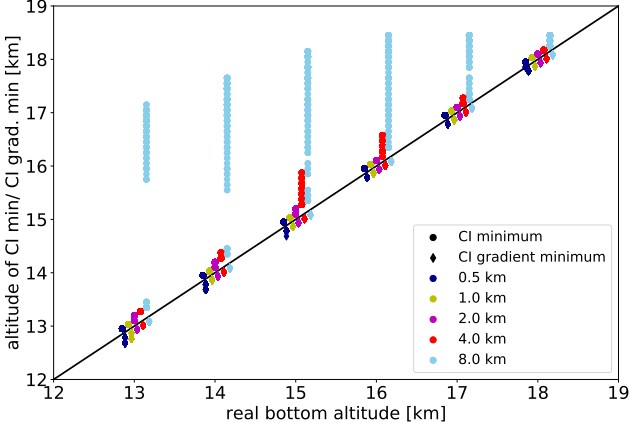

**Figure 8.** Altitude of CI minimum (full circles) and CI gradient minimum (full diamonds) against the real bottom altitude. The black line shows the line with slope 1. The different cloud thicknesses are shown colour coded and the points have been shifted along the line for clearness. Clouds with a CI minimum < 1.2 (optically thick) and > 5.0 (optically thin) have been excluded.

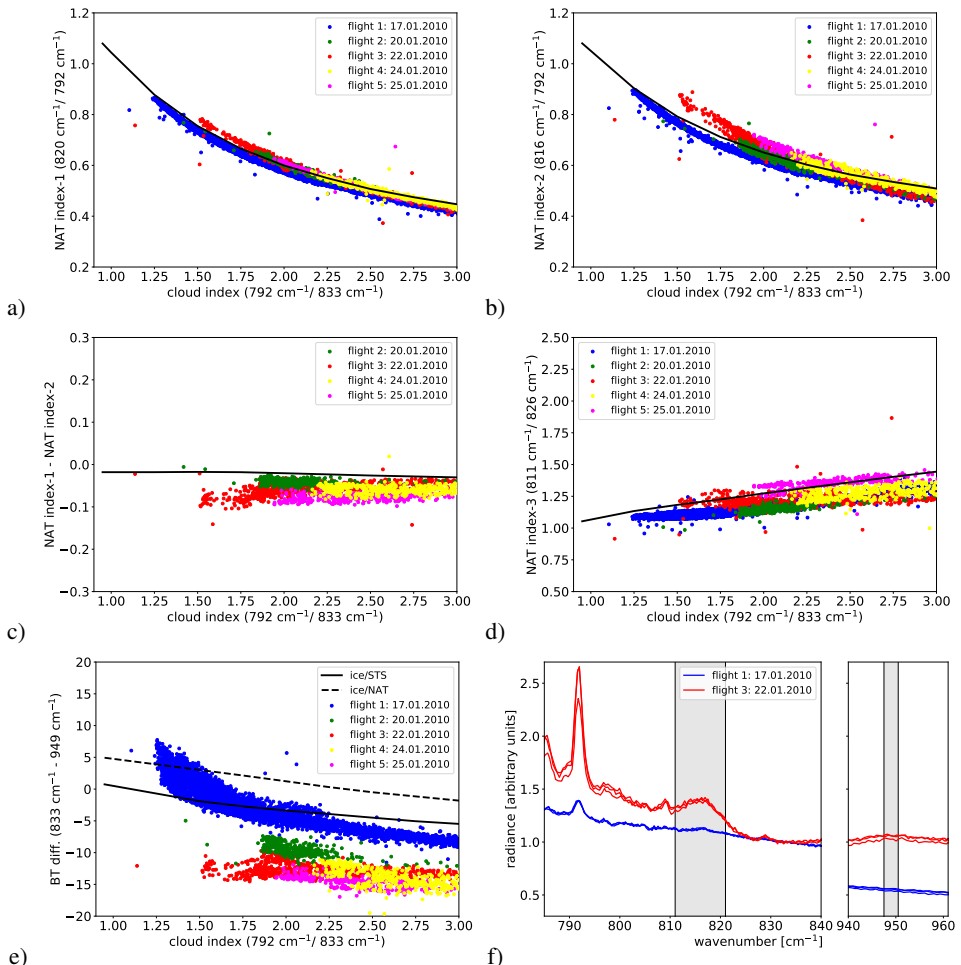

**Figure 9.** Correlations between the different NAT indices and cloud index for the CRISTA-NF observations during RECONCILE flights 1 – 5. a) NAT index-1 (819 – 821 $\mathrm{cm}^{-1}$)/(791 – 793 $\mathrm{cm}^{-1}$) (MW3/MW1); b) NAT index-2 (815 – 817 $\mathrm{cm}^{-1}$)/(791 – 793 $\mathrm{cm}^{-1}$) (MW4/MW1); c) NAT index-1 – NAT index-2; d) NAT index-3 (810 – 812 $\mathrm{cm}^{-1}$)/(825 – 827 $\mathrm{cm}^{-1}$) (MW5/MW6). The black lines show the separation lines, which mark the upper envelope of the regions of STS and ice (in a), b), and d)) or the region of medium and large NAT (in c)). Simulation results for ice and STS are in shown in dark and light blue, respectively. e) Correlation between the BT difference (832 – 834 $\mathrm{cm}^{-1}$) – (947.5 – 950.5 $\mathrm{cm}^{-1}$) and cloud index for the CRISTA-NF observations during RECONCILE flights 1 – 5. The black solid line shows the separation line between ice and STS and the dashed line marks the separation between ice and NAT. f) Example spectra from flight 1 and flight 3 for ice and NAT PSC, respectively. The spectra have been scaled such that the radiance for all spectra equals 1 in the spectral window 832.0 – 834.0 $\mathrm{cm}^{-1}$. The gray bars mark the region of the shifted NAT feature and the region used for the BTD around 949 $\mathrm{cm}^{-1}$.

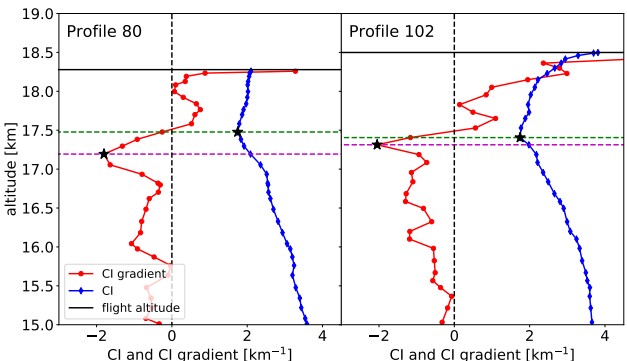

**Figure 10.** Altitude profiles of the cloud index (CI) and the CI gradient for two selected profiles of the CRISTA-NF measurements during RECONCILE flight 3. The CI is shown as a blue curve and the CI gradient as red curve. The flight altitude is marked by a horizontal black line. The dashed green and magenta lines show the minima of the CI and CI gradient profile, respectively.

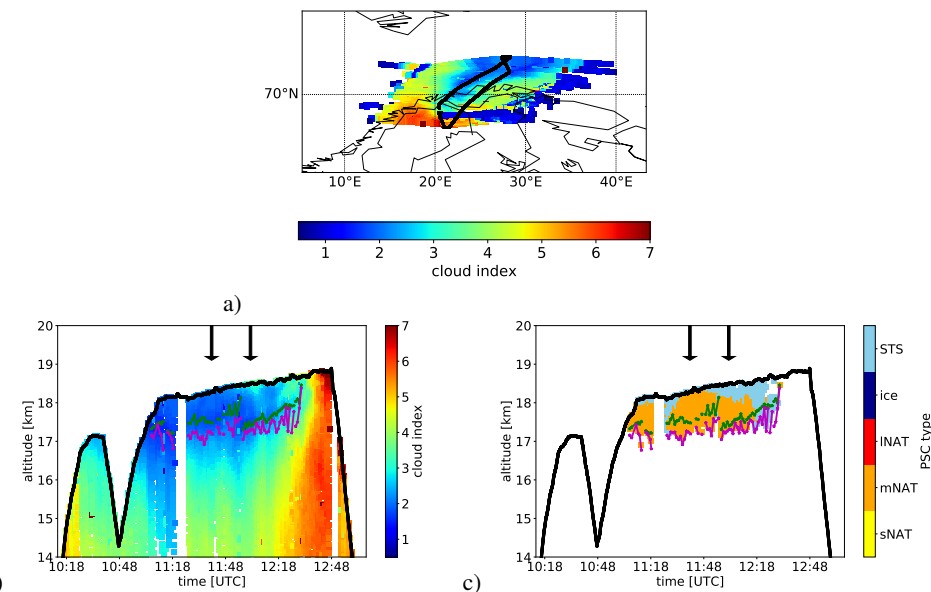

**Figure 11.** a), b) Longitude-latitude and cross section plot of the cloud index for RECONCILE flight 3. c) Cross section plot of the PSC types. The analysis of the PSC composition was performed for all spectra between flight altitude and CI gradient minimum. The black line shows the flight altitude of the aircraft. The CI minimum and the CI gradient minimum are marked by a green and a magenta line, respectively. Only profiles with a CI minimum below 3.0 and where both minima can sufficently by determined are considered. The black arrows in b) and c) mark the two profiles shown in Fig. 10.

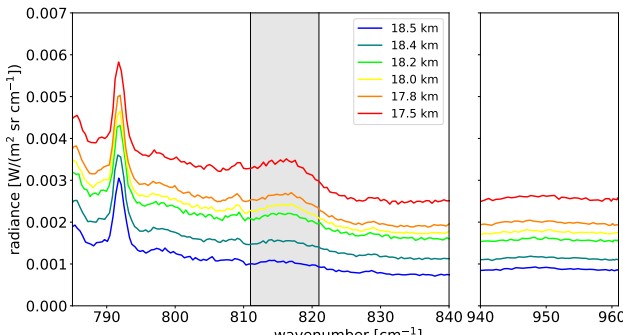

**Figure 12.** Example infrared spectra measured by CRISTA-NF during RECONCILE flight 3 inside PSCs at about 12:10 UTC. The spectra at different tangent altitudes are shown colour coded. The gray bar marks the region of the shifted NAT feature.

**Table 1.** CRISTA-NF instrument properties used in the simulations

| property | value |
|---|---|
| vertical sampling | 100 m |
| spectral sampling | $0.42 \text{ cm}^{-1}$ for $785 - 840 \text{ cm}^{-1}$ |
| | $0.59 \text{ cm}^{-1}$ for $940 - 965 \text{ cm}^{-1}$ |
| spectral resolving power $\frac{\lambda}{\Delta\lambda}$ | 536 at 12.5 μm |
| observer altitude | 18.4 km |
| altitude range | 10 km – observer altitude |

**Table 2.** Setup of the background atmosphere for the simulations

| constituent | source | spectral region |
|---|---|---|
| temperature | ERA-Interim (Dee et al., 2011) | both |
| pressure | ERA-Interim (Dee et al., 2011) | both |
| $CO_2$ | Diallo et al. (2017) | both |
| $HNO_3$ | climatology | both |
| $O_3$ | climatology | both |
| $ClONO_2$ | climatology | $785–840 \text{ cm}^{-1}$ |
| $H_2O$ | climatology | both |
| $HNO_4$ | climatology | $785–840 \text{ cm}^{-1}$ |
| $CCl_4$ | climatology | $785–840 \text{ cm}^{-1}$ |
| ClO | climatology | $785–840 \text{ cm}^{-1}$ |
| $NO_2$ | climatology | $785–840 \text{ cm}^{-1}$ |
| CFC-11 | climatology with update | both |
| | (using HAGAR (Riediger et al., 2000; Werner et al., 2010) and Bullister (2011)) | |
| HCFC-22 | climatology with update | $785–840 \text{ cm}^{-1}$ |
| | (using ACE-FTS (Boone et al., 2013)) | |
| CFC-113 | climatology with update | $785–840 \text{ cm}^{-1}$ |
| | (using Bullister (2011)) | |
| PAN | CRISTA-NF | both |
| $SF_6$ | climatology with update | $940–965 \text{ cm}^{-1}$ |
| | (using ACE-FTS (Boone et al., 2013)) | |
| $NH_3$ | climatology | $940–965 \text{ cm}^{-1}$ |
| $COF_2$ | climatology with update | $940–965 \text{ cm}^{-1}$ |
| | (using ACE-FTS (Boone et al., 2013) and Bullister (2011)) | |

**Table 3.** Cloud scenario simulation setup

| cloud dimension | values | | |
|---|---|---|---|
| PSC position | 13.0 – 30.0 km | | |
| PSC thickness | 0.5, 1.0, 2.0, 4.0, 8.0 km | | |

| PSC type | HNO$_3$ VMR [ppbv]* / volume density [µm$^3$/cm$^3$]** | radius [µm] | extinction [km$^{-1}$] |
|---|---|---|---|
| NAT | 1 – 15* | 0.5, 1.0, 1.5, 2.0, 2.5, 3.0, 3.5, 4.0, 5.0, 6.0, 8.0 | 3.4e$^{-5}$ – 2.2e$^{-3}$ |
| bimodal NAT | 3/7, 5/5, 7/3* (1st/2nd mode) | 1st mode: 0.5 – 2.5 2nd mode: larger than in 1st mode | 4.4e$^{-4}$ – 1.4e$^{-3}$ |
| STS with wt% H$_2$SO$_4$/HNO$_3$ 2/48, 25/25, and 48/2 | 0.1, 0.5, 1.0, 5.0, 10.0** | 0.1, 0.3, 0.5, 1.0 | 2.1e$^{-5}$ – 4.1e$^{-3}$ |
| NAT + STS wt% 2/48 | NAT: 5, 10, 15* STS: 5.0, 10.0** | NAT: 0.5 – 3.5 STS: 0.1, 0.3, 1.0 | 5e$^{-4}$ – 3.5e$^{-3}$ |
| ice | 0.1, 0.5, 1.0, 5.0, 10.0, 50.0, 100.0** | 1.0, 2.0, 3.0, 4.0, 5.0, 10.0 | 1.0e$^{-5}$ – 1.0e$^{-2}$ |

**Table 4.** Summary of the indices and their corresponding micro windows.

| name of index | definition | definition with short names |
|---|---|---|
| cloud index CI | (791 – 793 cm$^{-1}$)/(832 – 834 cm$^{-1}$) | (MW1)/(MW2) |
| NAT index-1 | (819 – 821 cm$^{-1}$)/(791 – 793 cm$^{-1}$) | (MW3)/(MW1) |
| NAT index-2 | (815 – 817 cm$^{-1}$)/(791 – 793 cm$^{-1}$) | (MW4)/(MW1) |
| NAT index-3 | (810 – 812 cm$^{-1}$)/(825 – 827 cm$^{-1}$) | (MW5)/(MW6) |
| BTD | BT(832 – 834 cm$^{-1}$) - BT(947.5 – 950.5 cm$^{-1}$) | BT(MW2) - BT(MW7) |