# Peer review of "A new method to detect and classify polar stratospheric NAT clouds derived from radiative transfer simulations and its first application to airborne IR limb emission observations"

_Atmospheric Measurement Techniques, 2020_

## Referee Comment (RC1) · Wolfgang Woiwode (Referee) · 6 Aug 2020

**General comments**

Based on radiative transfer simulations, the authors present a method to detect and discriminate PSCs in in airborne CRISTA-NF observations. NAT and ice are identified by using their characteristic spectral patterns. Detected PSCs, which do not show these signatures, are classified as STS. Assuming spectral characteristics of spherical particles, NAT PSCs are sub-grouped into small NAT (median $r \leq 1.0$ $\mu$m), medium-sized

NAT (median r=1.5 $\mu$m – 4.0 $\mu$m) and large NAT (median r$\geq$3.5 $\mu$m). Furthermore, a new method to detect the bottom altitude of non-opaque clouds is introduced. Application of the proposed methods to CRISTA-NF data is presented for one flight during the RECONCILE field campaign.

The presented study is interesting for PSC-related research, since aspects of PSC detection and classification are investigated in further detail and extended. The CRISTA-NF observations provide another interesting data set in the winter 2009/10, which was characterised by wide-spread PSC occurrence. The proposed approach to distinguish between different size classes of NAT particles might have potential to extend PSC classification using satellite instruments. However, the presented size classification is elaborated using model simulations only. Direct and detailed comparisons with observed spectra are not presented. Only spherical NAT particles are taken into account, while previous studies suggest that particle shape is an important issue. Comparisons with in situ observations could be helpful to test the proposed method. The method to determine cloud bottoms has the potential to gain more information on PSCs and thin cirrus clouds in the vertical domain. From my point of view, the manuscript is interesting and should be considered for publication in AMT after the following points have been addressed.

Main points:

1) From my point of view, it seems that the presented PSC classification and the size classification of NAT particles are elaborated separately from the observations. Simulated spectra in Figures 1 and 5 are presented in arbitrary units and are scaled individually, while only a few observations are shown separately in Figure 11 using radiance units. Furthermore, only a part of the used spectral range is shown for the observations (it seems that LRS 5 data is not shown). Therefore, the presented plots do not allow to evaluate whether the simulations reproduce details of the observations. From my point of view, this aspect is important, especially if issues related to particle shape are addressed. Overlays or residuals between simulated spectra and observations would

allow to evaluate to which degree the underlying assumptions of the classification are supported.

2) The proposed size classification is based on spectral characteristics of spherical NAT particles. However, field observations (Molleker et al. 2014) and laboratory experiments (Grothe et al., 2006) support a highly aspherical particle shape of large NAT particles or alternative habits. In a well-constrained scenario, the observed spectral fingerprint of large NAT particles has been shown to be clearly better compatible with highly aspherical particles (Woiwode et al., 2016). From my point of view, potential uncertainties of the proposed size classification related to the adopted particle shape should be discussed.

3) The spectral range used by the windows for NAT index-1 to -3 focuses only on the spectral region around the feature at ∼820 cmˆ-1. In previous work, it has been shown that, beside details of the spectral feature at ∼820 cmˆ-1, the overall spectrum and especially the spectral region towards 960 cmˆ-1 respond to particle size and shape (Woiwode et al., 2016). Are there certain (e.g. instrument-related) reasons for not considering the full available spectral range (and particularly the region 960 cmˆ-1) to constrain the size classification and define the criteria?

4) Previous studies showed the spectra of large NAT particles respond significantly to variable radiation from the troposphere/surface, which is scattered by the large particles into the line-of-sight (see Woiwode et al., 2016, 2019). The signal is modulated by the absence or presence of tropospheric clouds. To me it is not clear whether this aspect has been taken into account and which uncertainties might result in the size classification. (Were tropospheric clouds considered for the discussed flight?)

5) Application of the proposed classification scheme is presented for one flight and comparisons with in situ observations are not provided. In situ observations during the same flight suggest particle radii exceeding 7.5 and 10 $\mu$m (see Molleker et al., 2014, Table 1), while the proposed size classification suggests STS and medium-sized NAT

(r=1.5 $\mu$m – 4.0 $\mu$m) here. Are these results compatible? A comparison with in situ observations and including further flights could be helpful to evaluate (and potentially optimise) the proposed classification scheme.

6) Regarding the detection method for cloud bottoms, it would be interesting if the authors could discuss how the method responds if the cloud bottom is located above flight altitude and in the presence of several PSC layers located above each other. Such scenarios are frequently found in polar winters (e.g. Pitts et al., 2018).

Specific comments

L15, L492 It should be mentioned that the size classification assumes spherical NAT particles.

L35 References to further studies focusing on model simulations and comparisons with observations might be considered (e.g. Zhu et al., 2015; Zhu et al., 2017; Khosrawi et al., 2017, Tritscher et al., 2019).

L39ff Further PSC detection techniques (e.g. lidar, in situ) might be mentioned (e.g. Achtert and Tetsche, 2014; Molleker et al., 2014; Pitts et al., 2018).

L51 It should be noted that relatively high volume densities are required for this signature (see Höpfner et al., 2006, 2006b), which are found less frequently in the Arctic.

L67, L227 "Very small" should be defined.

L67 "Another spectral feature" should be explained. To me it seems to be the same spectral signature of beta-NAT particles reported by previous studies, which is modulated by the actual PSC scenario, particle size distribution, particle shape and scattered light from below. "A similar spectral feature" seems more appropriate to me.

L70 It should be mentioned that also other particle modes between 1 $\mu$m and 6 $\mu$m were analysed by the same study (see Woiwode et al., 2016, Table 2, Fig. 10ff, Appendix B).

L72 It should be mentioned that particle shape and scattered radiation from below where also found to be important parameters by the same study.

L127ff Information on the along-track sampling would be helpful.

L167 The overall cloud scenario used is well-described. However, the tropospheric cloud scenario and implications for scattered radiation from below, which is scattered by the particles, are not addressed (see 4)).

L163 Did the simulations account for CFC-12? (spectral band centered at ∼920 cmˆ-1)

L202, L222 Detailed analysis in the Woiwode et al. (2016) showed that the shape of the feature is sensitive to particle shape.

L230ff See 1)-4): The observed spectral fingerprint is not exploited fully to define of criteria of the classification. E.g. the spectral region around 960 cmˆ-1, which was shown to respond significantly to size distributions and particle shape (see Woiwode et al., 2016) is not considered. It is not clear to me whether modulation of the spectra by scattered light from the below is considered. Direct comparisons between observed and simulated spectra are not presented to verify the underlying assumptions.

L279, L314, L336, L507 It should be noted that these statements on the performance are valid within the assumptions made. Verification by direct comparisons of simulated and observed spectra and comparisons with other observations are not provided here.

L315ff For comparisons with other studies, it would be helpful to include a Figure showing the used size distributions.

L383ff See 6): It would be interesting if the authors could discuss how the method responds if the cloud bottom is located above flight altitude and in the presence of several PSC layers located above each other. Such conditions are frequently found in polar winters (e.g. Pitts et al., 2018).

L392 The motivation for choosing the discussed flight should be provided.

L397 It would be helpful to indicate the location of the selected two profiles in Fig. 10 and comment on the composition derived in Fig. 10b.

L416ff See 5): Considering one case study constitutes only a limited test for a new classification scheme. Further RECONCILE PSC flights, collocated in situ observations and e.g. CALIPSO observations could be used to test (and potentially optimise) the proposed classification scheme.

L422 Why are only spectra with CI < 3.0 shown here? (compare Fig. 3 and L388)

L424 See 1): Figure 11 shows examples of PSC spectra in radiance units, while Figures 1 and 5 show simulations in arbitrary units, which are individually scaled. Clear comparisons between simulations and observations are not possible. Only a part of the used spectral range is shown for the observations (compare L99). Overlays or residuals between simulated and observed spectra using the same units would be helpful to evaluate to which degree the simulations meet details of the observations.

L435,436 See 5): Here, the CRISTA-NF observations are classified as STS or medium sized (median r=1.5-4.0 $\mu$m) NAT. However, collocated in situ observations suggest particles with radii exceeding 7.5 and 10 $\mu$m (see Molleker et al., 2014, Table 1). Are these results compatible? It would be interesting to compare size distributions, which are supported by the CRISTA-NF simulations, with the in situ observations.

L472 The conclusion on the spectral range from 833 to 960 cm-1 is not supported sufficiently by Fig. 5, since the spectral range from 840 to 940 cm-1 is not shown.

L474ff From my point of view, the conclusion "very similar" is not sufficiently supported. The Woiwode et al. (2016,2019) studies used a broader spectral range, higher spectral resolution and spectra in absolute radiance units (which constitutes another "piece of information"). Furthermore, these studies used direct overlays and residuals of simulated observed spectra to analyse the spectral fingerprint of large NAT particles in detail and to define criteria for detection. In the study presented here, the full available spectral range is not exploited and no direct comparisons between simulated and observed spectra are provided.

However, within the abovementioned limitations, the conclusion on similarity is supported to some degree by the Woiwode et al. (2016) study, Appendix B, for the small and medium particle sizes. In this study it was shown that for small NAT particles (r=1 $\mu$m), spherical and aspherical particle populations show an almost identical spectrum, while at r=3 $\mu$m the signatures start to diverge notably. Furthermore, one might speculate that particles become gradually more aspheric when growing to large sizes, and that (nearly) spherical particles are a suitable assumption at earlier growing stages.

L476ff It has been shown before that a similar signature can be simulated by assuming spherical particles (see Woiwode et al., 2016, Appendix B). Furthermore it should be noted that the possibility of some variability in the NAT phase regarding particle habits has been clearly mentioned in previous studies (Molleker et al., 2014; Woiwode et al., 2016) and has not been ruled out in the Woiwode et al. (2019) study.

However, using the combination of a wide spectral range, high spectral resolution, and supporting information from in situ observations, the Woiwode et al. (2016,2019) studies were not successful in reproducing details of developed spectral fingerprints of large NAT particles by assuming spherical particles. The observed combination of a strong "step-like" feature and a flat spectral baseline of the observed spectra towards higher wave numbers (described by the simplified "hockey-stick" picture in Woiwode et al., 2019) could not be reproduced. Close inspection of the spectra showed that Mie simulations of spherical particles always showed significant differences from the observed "shifted feature" around 820 cmˆ-1 and a significant negative slope and/or an "upward arching" of the spectral baseline at higher wave numbers, and further discrepancies. These discrepancies resulted in significant patterns in the residuals. Using spectral characteristics of highly aspherical particles clearly improved the residuals. Thus, the spectrum clearly includes information on particle shape. Also from other work it is known that infrared spectroscopy allows to infer information on particle shape (Wag-
ner et al., 2005). However, it is understandable to me that within the limited spectral range considered here and without detailed comparisons between observations and simulations a clear decision appears to be not feasible here.

From my point of view, the question is not only whether any "hockey-stick" signature can be modelled, but also whether details of the entire accessible spectral fingerprint in developed signatures of large NAT particles can be reproduced, and how the results compare with other observations.

L477f This sentence should be revisited, since this has been done in before (Woiwode et al., 2016,2019). Of course, any further case studies including in situ comparisons would be helpful to further constrain properties of large NAT particles.

L480f Here, clarification is required. Since absolute radiances are shown in Figure 11, I would expect that integrated radiances and their differences can be calculated.

L484 See comment to L476ff: it has been shown before that a similar signature can be simulated by assuming spherical particles. However, close inspection showed significant discrepancies between simulated and observed spectra in the case of spherical particles.

Fig 11 The channel LRS 5 seems not to be shown (compare L99).

Fig 10 An additional panel including a map with the geolocations of the observations would be helpful to visualize the location and size of the sampled region.

Technical corrections

L32 denitrifaction -> denitrification

L115 KOPRA should be expanded

L450 CALIPSO and CALIOP should be expanded and a reference should be provided (e.g. Pitts et al., 2018)

L458 transfered -> transferred

L478 FSSP should be expanded and a reference should be provided

L500 GLORIA should be expanded and a reference should be provided

L508 Sentence should be revisited. How can new PSC observations be obtained from the presented method? (Possibly, "observations" should be replaced e.g. by "data set")

Fig 3, Fig 6 Plots may be refined: data points seem to overlap strongly and it is not clear whether significant populations of data points are hidden below other data points. At least, this should be discussed in the text.

Caption Fig 4 spetrum -> spectrum

Fig 7b The x-axis might be scaled from e.g. -4 to >8 for a better focus on the relevant region

Literature

Achtert, P. and Tesche, M.: Assessing lidar-based classification schemes for polar stratospheric clouds based on 16 years of measurements at Esrange, Sweden, Journal of Geophysical Research: Atmospheres, 119, 1386–1405, https://doi.org/10.1002/2013jd020355, 2014

Grothe, H., Tizek, H., Waller, D., and Stokes, D. J.: The crystallization kinetics and morphology of nitric acid trihydrate, Phys. Chem. Chem. Phys., 8, 2232–2239, doi:10.1039/B601514J, 2006

Höpfner, M., Larsen, N., Spang, R., Luo, B. P., Ma, J., Svendsen, S. H., Eckermann, S. D., Knudsen, B., Massoli, P., Cairo, F., Stiller, G., v. Clarmann, T., and Fischer, H.: MIPAS detects Antarctic stratospheric belt of NAT PSCs caused by mountain waves, Atmos. Chem. Phys., 6, 1221–1230, doi:10.5194/acp-6-1221-2006, 2006b

Khosrawi, F., Kirner, O., Sinnhuber, B.-M., Johansson, S., Höpfner, M., Santee, M. L.,

Froidevaux, L., Ungermann, J., Ruhnke, R., Woiwode, W., Oelhaf, H., and Braesicke, P., Denitrification, dehydration and ozone loss during the 2015/2016 Arctic winter, Atmos. Chem. Phys., 17, 12893-12910, https://doi.org/10.5194/acp-17-12893-2017, 2017

Pitts, M. C., Poole, L. R., and Gonzalez, R.: Polar stratospheric cloud climatology based on CALIPSO spaceborne lidar measurements from 2006 to 2017, Atmos. Chem. Phys., 18, 10 881–10 913, https://doi.org/10.5194/acp-18-10881-2018, 2018

Tritscher, I., Grooß, J.-U., Spang, R., Pitts, M. C., Poole, L. R., Müller, R., and Riese, M., Lagrangian simulation of ice particles and resulting dehydration in the polar winter stratosphere, Atmos. Chem. Phys., 19, 543-563, https://doi.org/10.5194/acp-19-543-2019, 2019

Wagner, R., Möhler, O., Saathoff, H., Stetzer, O., and Schurath, U.: Infrared spectrum of nitric acid dihydrate - Influence of particle shape, J. Phys. Chem. A, 109, 2572–2581, 2005.

Zhu, Y., Toon, O. B., Lambert, A., Kinnison, D. E., Brakebusch, M., Bardeen, C. G., Mills, M. J., and English, J. M., Development of a Polar Stratospheric Cloud Model within the Community Earth System Model using constraints on Type I PSCs from the 2010–2011 Arctic winter, J. Adv. Model. Earth Sy., 7, 551–585, https://doi.org/10.1002/2015MS000427, 2015

Zhu, Y., Toon, O. B., Pitts, M. C., Lambert, A., Bardeen, C., and Kinnison, D. E., Comparing simulated PSC optical properties with CALIPSO observations during the 2010 Antarctic winter, J. Geophys. Res. A, 122, 1175–1202, https://doi.org/10.1002/2016JD025191, 2016JD025191, 2017
* * *

---

## Referee Comment (RC2) · Anonymous Referee #2 · 24 Aug 2020

The paper describes methods to detect PSCs and to classify them. The following PSC types are distinguished: NAT (nitric acid trihydtrate), STS (supercooled ternary solution) and ice clouds. For the detection of small NAT particles, a spectral feature at about 820cm-1 has often been used. The authors show, that for larger NAT particles this feature shifts towards smaller wavenumbers and they use this feature to further specify the NAT particles as sNAT (small particles), mNAT (medium particles), and lNAT (large particles). Ice is detected by using the difference between 2 spectral regions (∼833cm-1 and ∼950cm-1), the radiance is similar in clear sky conditions but

significantly smaller at 950cm-1 when ice is present, a well-known feature which has already been used before. When the cloud is neither NAT nor ice, it is classified as STS. Further a method based on the so-called Cloud Index (CI) is developed to derive the cloud bottom height.

The methods are applied to a comprehensive set of radiative transfer simulations including NAT, ice and STS, all with various concentrations and particle size distributions. Based on the simulations it is demonstrated that the methods work well, in particular sNAT can be distinguished very well from the larger NAT particles mNAT and lNAT. Finally the new methods are also applied to observations of the CRISTA-NF instrument taken during a flight of the RECONCILE campaign.

Generally the newly developed methods are described quite well and the figures are appropriate. However, I think that some clarifications are needed, in particular the motivation is not so clear for readers who do not work in the specific field of research (see also my comments below). The scope of the paper fits AMT very well, therefore I recommend publication after taking into account the comments below.

General comments:

* The motivation is not clear. It is said in the abstract that there are "uncertainties in the representation of PSCs in model simulations". Which models? Atmospheric chemistry models, climate models? Please provide some examples of models which can be improved by better knowledge of PSC properties. Which properties are required by these models? What are the most relevant parameters? Composition, concentration, size, shape ...? The methods presented here do not derive number concentrations, can these also be derived from spectral IR observations? How important is the classification into different size regimes?

* Cloud optical thickness is a useful parameter to describe a cloud, the term is sometimes used in the paper but no number is given. What is the optical thickness range of the clouds considered here (in the RT simulations and what can be observed using the

limb observations)? I suppose that all clouds are optically very thin, what is the upper limit of cloud optical thickness that can be analysed with the presented methods?

* The radiative transfer simulations are based on a single scattering approach which is not described in detail here. This also means that it is only valid for very thin clouds, for which multiple scattering can be neglected (see e.g. Höpfner and Emde 2005). Please provide the optical thickness range used for the radiative transfer simulations to justify the neglect of multiple scattering.

* Presuambely, the JURASSIC model does not account for horizontal inhomogeneities. Please discuss the validity of this approach for PSCs, i.e. how large is the horizontal extend of the PSCs typically compared to the line of sight through the PSCs?

* For the reader it is rather difficult to remember the definition of all indices used in the paper. I think it would be helpful to include a figure showing the spectral regions used and mark the spectral windows that are used to calculate the individual indices. Also a table including the definitions of NAT-indices 1,2,3 and the CI index could be useful.

* I miss some discussion about the uncertainties. For example on p.8 you write: "Nearly all simulations with NAT particles < 3 $\mu$m lie above the region of the simulations for STS and ice clouds, which is marked by the solid black line. Thus, NAT particles within this size range can be detected and discriminated using NAT index-1." For the modelled spectra this is correct, but also for real observations? Measurements always include uncertainty, how accurate measurements are required so that NAT is clearly separated from the ice clouds?

Specific comments:

* The title is a bit misleading. The main focus of the paper are the methods to classify and detect PSCs. I thought from reading the title starting with "Radiative transfer" that the paper was more about radiative transfer methods etc. For the RT simulations a well-known model JURASSIC is applied but the methodology is not described in this

paper. Also the observational methods are not described here. I think that the title should be something like e.g. "A new method to detect and classify polar stratospheric clouds"

* Abstract: p1, l7 "... showed a spectral peak at about 816 cm-1 . This peak is shifted compared to the peak at about 820 cm-1, which is known to be caused by small NAT particles. " -> A bit more information about this peak would be helpful. What is the physical process responsible for the peak. Which physical processes could produce a shift of a spectral peak ...

p1, l16: "gradient of the CI" -> which gradient is meant here? -> "vertical gradient"

p3, l31-33: These 3 sentences should be shifted to Section 2.2

p4. l19: Title of section should include the term "radiative transfer simulations"

p4 l32: "The optical properties of the particles, extinction coefficient, scattering coefficient, and phase function, required for the radiative transfer simulations ..." -> "The optical properties of the particles (extinction coefficient, scattering coefficient, and phase function) required for the radiative transfer simulations ...", include brackets here because extinction coefficient, scattering coefficient, and phase function are the optical properties

p7 l20: What is the "scattering radius" of a particle, this term has not been defined

p7 l31: "different contributions of extinction and scattering" -> extinction is the sum of absorption and scattering, therefore I think that you mean "absorption and scattering"

Fig.1 and 5: "spectra have been scaled such that the radiance equals 1" -> scaling factor is not clear, is it the average radiance over the plotted spectral range?

Fig.2: For the interpretation of the RT simulations, it would help to include here also the refractive indices of ice and STS. Further I think that it would be very helpful to show extinction and absorption coefficients, which are probably calculated using Mie theory

for the individual PSC types and for some particle sizes to see, that for larger particles the scattering coefficient dominates.

p9, l4: "results for PSCs with larger NAT particles (up to 4 $\mu$m) also lie above the simulations for STS and ice (black separation line)": In Fig.3 it is not well visible, which radii lie above the separation line. May be discrete colours could be used for each of the simulated radii?

p9, l23: "detected NAT spectra" -> NAT is detected, not the "spectra"

p10, l23: "When the NAT detection and classification procedure (described in the previous subsection) is applied to the simulation results for the mixed clouds, the good discrimination between the small and medium size particles remains." -> this is not shown, why?

p13, l3-9: Here you discuss about "cloud optical thickness" (thick and thin) without providing any values -> as mentioned above, please quantify the cloud optical thickness.

p15, l7: "... where much more NAT was observed by CALIOP (the difference in the Southern hemisphere is much smaller)"-> how much more NAT was observed by CALIOP, how much smaller is the difference in the Southern hemisphere?

p16 l20: "This method can surely be transferred to other cloud observations such as cirrus clouds and aerosol layers and to other airborne instruments measuring in the same wavelength region like e.g. GLORIA." -> Which clouds and aerosols could be observed? Probably only very thin clouds? Up to which cloud optical thickness can the method be applied?

Reference:

* M. Hoepfner and C. Emde. Comparison of single and multiple scattering approaches for the simulation of limb-emission observations in the mid-IR. J. Quant. Spectrosc. Radiat. Transfer, 91(3):275-285, March 2005.

---

## Referee Comment (RC3) · Anonymous Referee #3 · 25 Aug 2020

Review of amt-2020-144:

"Radiative transfer simulations and observations of infrared spectra in the presence of polar stratospheric clouds: Detection and discrimination of cloud types" by Kalicinsky et al.

This paper demonstrates the clear capability of infrared FTS limb sounders to provide detection, discrimination of particle types and particle sizing in polar stratospheric clouds and is an advance on the current state of the art. The paper is acceptable for

publication following some minor corrections.

I strongly suggest that an attempt is made to make an additional plot that shows the optical depth vs CI for some samples of different PSC cloud types. Likewise the CI vertical gradient is related to the optical thickness gradient.

Specific comments and typos

[xxx] means add xxx /xxx/ means delete xxx

Page2

L35: "incomplete" rather than "difficult"?

L37-38: [An] infrared .... build[s]...

L39: sounder[s]

L44-45: make distinction between CO2 molecular line emssion and broader continuum like aerosol emission?

L51: color => colour

L64-65: Maybe make clearer that for an airborne instrument the limb tangent moves away from the aircraft for downward looking views.

L72: What about changes with the aspect ration of the particles?

L86: itselves => themselves

L87: JURASSIC is not defined until L106

L110: PREMIER IRLS is not defined

L111: spectral/ly/

L115: KOPRA is not defined

L130: What about spectral regions for STS and background binary sulfate aerosols?

L133: al[t]itude

L139: reference to a rejected ACPD paper?

L177: median radius varied in steps of?

L215: imaginary part/s/

L234-249 and everywhere else including figure captions and tables: Is it possible to give all these spectral regions a distinct short name? e.g. R1, R2, R3 etc Otherwise the reader has to scan the characters and check to see which regions are the same thing rather than seeing that immediately from the short name.

L257: an[d]

L312: less => fewer

L335: mille => thousand

L408: extend => extent

L463: called [a] hockey-stick

L494: What are the detection levels of the new method compared to the old method? e.g. in terms of the minimum volume density um3/cm3.

L501: "minimisation" means "reduction"?

L507: "safely" means "always"?

Figures 3, 6 and 7: Can an approximate optical depth scale be put on the x-axis?

Figure 8: What are the actual optical depths corresponding to these CI values?

---

## Author Comment (AC1) · 26 Oct 2020

Reply to the comments of reviewer 1 on the manuscript

Radiative transfer simulations and observations of infrared spectra in the presence of polar stratospheric clouds: Detection and discrimination of cloud types

by C.Kalicinsky et al.

We thank the reviewer for the helpful comments and recommendations. In the following, we discuss the issues addressed by the reviewers and explain our opinions and the modifications of our manuscript.

We enumerate the comments and repeat them in bold face. The modifications of the manuscript are displayed in the marked-up manuscript version as colored text. Deleted parts are shown in red and new or modified text parts in blue.

**1 Comments**

**General comments**
**Based on radiative transfer simulations, the authors present a method to detect and discriminate PSCs in airborne CRISTA-NF observations. NAT and ice are identified by using their characteristic spectral patterns. Detected PSCs, which do not show these signatures, are classified as STS. Assuming spectral characteristics of spherical particles, NAT PSCs are sub-grouped into small NAT (median r≤1.0$\mu$m), medium-sized NAT (median r=1.5$\mu$m − 4.0$\mu$m) and large NAT (median r≥3.5$\mu$m). Furthermore, a new method to detect the bottom altitude of non-opaque clouds is introduced. Application of the proposed methods to CRISTA-NF data is presented for one flight during the RECONCILE field campaign.**
**The presented study is interesting for PSC-related research, since aspects of PSC detection and classification are investigated in further detail and extended. The CRISTA-NF observations provide another interesting data set in the winter 2009/10, which was characterised by wide-spread PSC occurrence. The proposed approach to distinguish between different size classes of NAT particles might have potential to extend PSC classification using satellite instruments. However, the presented size classification is elaborated using model simulations only. Direct and detailed comparisons with observed spectra are not presented. Only spherical NAT particles are taken into account, while previous studies suggest that particle shape is an important issue. Comparisons with in situ observations could be helpful to test the proposed method. The method to determine cloud bottoms has the potential to gain more information on PSCs and thin cirrus clouds in the vertical domain. From my point of view, the manuscript is interesting and should be considered for publication in AMT after the following points have been addressed.**
**Main points:**

1. **From my point of view, it seems that the presented PSC classification and the size classification of NAT particles are elaborated separately from the observations. Simulated spectra in Figures 1 and 5 are presented in arbitrary**

**units and are scaled individually, while only a few observations are shown separately in Figure 11 using radiance units. Furthermore, only a part of the used spectral range is shown for the observations (it seems that LRS 5 data is not shown). Therefore, the presented plots do not allow to evaluate whether the simulations reproduce details of the observations. From my point of view, this aspect is important, especially if issues related to particle shape are addressed. Overlays or residuals between simulated spectra and observations would allow to evaluate to which degree the underlying assumptions of the classification are supported.**

It is correct, our classification method is derived from simulated NAT, STS, and ice PSC spectra. Our intention was to use a wide spread of different situations to stake out the possible range of signals due to variations in the PSC particle size distribution (as done before by Höpfner et al (2006), Spang et al. (2012,2016)). As our detection and classification method is based on relative effects, i.e. colour ratios and brightness temperature differences, we can show where the CRISTA-NF measurements are located in the color ratio and BTD range, without fitting the absolute radiance of each single spectrum. Scaling the spectra to 1 at $833\,\text{cm}^{-1}$ (Figs. 1,2,5) helps to point out the relative differences that our method relies on.

We added the second CRISTA-NF channel. We included new figures in the manuscript that show the correlations between the NAT indices/ the BTD and the CI for the observations of CRISTA-NF (Fig. 9) to allow for a comparison with the simulation results (Figs. 3,6). We also included data from additional flights of the RECONCILE campaign to demonstrate that the broad range of simulations covers well the range of observations (Fig. 9).

All results are shown in the new subsection 4.1.

2. **The proposed size classification is based on spectral characteristics of spherical NAT particles. However, field observations (Molleker et al. 2014) and laboratory experiments (Grothe et al., 2006) support a highly aspherical particle shape of large NAT particles or alternative habits. In a well-constrained scenario, the observed spectral fingerprint of large NAT particles has been shown to be clearly better compatible with highly aspherical particles (Woiwode et al., 2016). From my point of view, potential uncertainties of the proposed size classification related to the adopted particle shape should be discussed.**

The study is based on simulations for spherical particles as this particle shape can be simulated with less computational effort compared to aspherical particles, which is an important factor with respect to the large number of simulations that were carried out. We did not question the existence of aspherical NAT. However, our simulation results demonstrate that spherical particles can produce all the different appearances of the NAT feature (typical feature, shifted feature, step-like behaviour) including those previously attributed to highly aspherical NAT. We think that this is an important point to consider as it implies that IR limb emission measurements alone are not sufficient to derive the existence of aspherical NAT, in particular for cases without additional information on the atmospheric state and cloud particle size distribution.

Our approach of grouping the NAT particles into 3 size bins based on the spectral signature we do not consider affected by the particle shape for the following reasons. The

small particles ($< 3~\mu$m) are considered not or only slightly aspherical. In our study we used the term 'small' for particle size distributions with median radii $\leq 1.0~\mu$m. Also Woiwode et al. (2019) use the method by Höpfner et al. (2006), which is based on simulations using spherical particles, to detect these small particles. A substantial fraction of our 'medium' size group (median radii $\leq 4~\mu$m) can also be considered nearly spherical. The shifted NAT signature reported by Woiwode et al. (2016) for highly ashperical NAT is bi-modal with median radii of 2 and $4.8~\mu$m, where the latter is dominating the signal. This means that the shifted NAT signature that so far was solely attributed to the asphericity, has only been observed in the presence of larger NAT particles. Further, Woiwode et al. (2019, Fig. 1) show that for median radii $\geq 5.0~\mu$m a step-like NAT-feature can be observed. Hence, our result that the shift of the NAT signature is associated with NAT particle size is in line with the results of Woiwode et al. (2016, 2019). Due to particle asphericity our size bin boundary between 'medium' and 'large' has uncertainties. We added these considerations and the contraint that our size bin boundaries are only valid under the assumption of spherical particles to the discussion.

A detailed comparison between the effects of aspherical, mono-modal , and bi-modal spherical cases is beyond the scope of the paper and should be subject of a future study.

3. **The spectral range used by the windows for NAT index-1 to -3 focuses only on the spectral region around the feature at $\sim$820 cm$^{-1}$. In previous work, it has been shown that, beside details of the spectral feature at $\sim$820 cm$^{-1}$, the overall spectrum and especially the spectral region towards 960 cm$^{-1}$ respond to particle size and shape (Woiwode et al., 2016). Are there certain (e.g. instrument-related) reasons for not considering the full available spectral range (and particularly the region 960 cm$^{-1}$) to constrain the size classification and define the criteria?**

   Our detection method is able to detect nearly all simulated NAT scenarios by using the spectral range between 810 and 834 cm$^{-1}$. It misses only a small part of the very large particles with radii of 8 $\mu$m (see Fig. 4 b)). The gradient between 834 cm$^{-1}$ and 950 cm$^{-1}$ is similar for NAT and STS spectra as shown in Fig. 6. We used 950 cm$^{-1}$ instead of 960 cm$^{-1}$ to avoid CRISTA-NF measurement artefacts at the boundary of this channel, but checked in the simulations that both windows deliver similar results. Thus, adding the 950 cm$^{-1}$ spectral region to the classification does not improve the discrimination between NAT and STS.

   Moreover, the simulations for NAT/STS mixtures and bi-modal NAT showed that the region around 950 cm$^{-1}$ is very sensitive to these combinations (see Fig. 5). In our opinion there are multiple ways to obtain similar radiance values in this region. By contrast to this, the spectral feature in the region $810 - 820$ cm$^{-1}$ always gives information on the particle type and size, even in the case of the mixtures.

4. **Previous studies showed the spectra of large NAT particles respond significantly to variable radiation from the troposphere/surface, which is scattered by the large particles into the line-of-sight (see Woiwode et al., 2016, 2019). The signal is modulated by the absence or presence of tropospheric clouds. To me it is not clear whether this aspect has been taken into account and which uncertainties might result in the size classification. (Were tropo-**

**spheric clouds considered for the discussed flight?**

The influence of tropospheric clouds was intensively studied during the analysis of the radiative transfer simulations for various PSC situations for the MIPAS-Env satellite instrument (Spang et al., 2012, 2016) and the results can be transfered to here. A tropospheric cloud below the PSC mainly influences the absolute radiance values where the cold tropospheric cloud leads to lower overall radiance values. The appearance of the spectral signature (typical peak, shifted peak, or step-like behaviour) in the region $810$ cm$^{-1}$ to $820$ cm$^{-1}$ is nearly unchanged. Simulations by Woiwode et al. (2019) also show this. As our detection method is based on colour ratios, i.e. relative effects, the change of the absolute radiance values does not significantly influence the analysis. Especially in the case of STS or ice, which define the separation lines (see Fig. 3), the data points in the scatter plots only slightly shift along the correlation lines/regions for the specific PSC type. Thus, the separation lines are still valid in such cases.

We added a new subsection 3.2.2 to the manuscript.

5. **Application of the proposed classification scheme is presented for one flight and comparisons with in situ observations are not provided. In situ observations during the same flight suggest particle radii exceeding 7.5 and 10$\mu$m (see Molleker et al., 2014,Table 1), while the proposed size classification suggests STS and medium-sized NAT (r=1.5$\mu$m − 4.0$\mu$m) here. Are these results compatible? A comparison with in situ observations and including further flights could be helpful to evaluate (and potentially optimise) the proposed classification scheme.**

We applied the proposed classification scheme to all PSC flights (1-5) during RECONCILE. The results are now presented in Figure 9 and discussed in Section 4.1.

Unfortunatelly, the number of particles with radii $> 7.5\mu$m (59 particles in the whole flight) is the only information about the size distribution during flight 3 (22.01.2010). Without any relation to the number of small particles a serious comment on the agreement between CRISTA-NF and those measurements cannot be made. Grooß et al. (2014) present a size distribution for a part of this flight, which shows that most particle radii lie in the range < about 6 $\mu$m with a maximum at about 2.5/3 $\mu$m. This agrees with our results. The same is true for flight 4 (shown in Molleker et al. (2014)). However, we refrain from such a simple comparison between in situ observations and our results as there are too many differences between the instruments. Since CRISTA-NF observes all air masses along the line of sight (up to hundreds kilometres away from the aircraft), the air masses detected by the in situ instruments may play only a minor role for the infrared spectra.

6. **Regarding the detection method for cloud bottoms, it would be interesting if the authors could discuss how the method responds if the cloud bottom is located above flight altitude and in the presence of several PSC layers located above each other. Such scenarios are frequently found in polar winters (e.g. Pitts et al., 2018).**

When the PSC is located above the flight altitude the CI typically increases with decreasing altitude from flight altitude downwards, because the length of the line of sight inside the PSC decreases also. Thus, there is not the typical large increase of the CI indicated by the minimum of the CI gradient when leaving the cloud. Both, the CI

and the CI gradient, typically show the lowest values at the points closest to the flight altitude.

In the case of layered PSCs, where PSCs of different type are lying above each other, the results depend on the optical thickness of the different layers. On the one hand, the CI gradient minimum is still located at the bottom of the complete PSC, when the lower layer is optically thick enough and the transition to the cloud free atmosphere shows the largest increase in CI. On the other hand, when the lower layer is optically very thin, it can occur that the CI gradient minimum is located directly below the upper layer. In this case the transition from the upper layer to the lower layer leads to a larger increase in CI than the transition from the lower layer to the cloud free atmosphere.

We added these information to section 3.5.

**Specific comments:**

1. **L15, L492 It should be mentioned that the size classification assumes spherical NAT particles.**
   We added the information that the classification is based on spherical particles.

2. **L35 References to further studies focusing on model simulations and comparisons with observations might be considered (e.g. Zhu et al., 2015; Zhu et al., 2017; Khosrawi etal., 2017, Tritscher et al., 2019).**
   We added additional information. See Reviewer 2.

3. **L39ff Further PSC detection techniques (e.g. lidar, in situ) might be mentioned (e.g. Achtert and Tetsche, 2014; Molleker et al., 2014; Pitts et al., 2018).**
   We added information on other measurement techniques.

4. **L51 It should be noted that relatively high volume densities are required for this signature (see Höpfner et al., 2006, 2006b), which are found less frequently in the Arctic.**
   We added this information.

5. **L67, L227 "Very small" should be defined.**
   The sizes that are necessary to shift the spectral feature are evaluated in this paper and the numbers are given later in Sec. 3. Thus, we do not add information in the introduction.
   In the case of the BTD only the particles $< 1.0$ $\mu$m lead to a large radiance decrease between the two micro windows. We added this information.

6. **L67 "Another spectral feature" should be explained. To me it seems to be the same spectral signature of beta-NAT particles reported by previous studies, which is modulated by the actual PSC scenario, particle size distribution, particle shape and scattered light from below. "A similar spectral feature" seems more appropriate to me.**
   It is a spectral signature of beta-NAT, but in the first cases it is a peak and here a step-like feature. Thus, it is another appearance of the beta-NAT signature. We corrected this.

7. **L70 It should be mentioned that also other particle modes between $1\mu$m and $6\mu$m were analysed by the same study (see Woiwode et al., 2016, Table 2, Fig. 10ff, Appendix B).**
   We added this information.

8. **L72 It should be mentioned that particle shape and scattered radiation from below where also found to be important parameters by the same study.**
   We added this information.

9. **L127ff Information on the along-track sampling would be helpful.**
   We added this information to Sect. 2.1 as this section describes the CRISTA-NF instrument.

10. **L167 The overall cloud scenario used is well-described. However, the tropospheric cloud scenario and implications for scattered radiation from below, which is scattered by the particles, are not addressed (see 4)).**
    See general comments 4.

11. **L163 Did the simulations account for CFC-12? (spectral band centered at $\sim$920 cm$^{-1}$).**
    No, because the influence of CFC-12 at 940 cm$^{-1}$ or larger wave numbers is insignificant.

12. **L202, L222 Detailed analysis in the Woiwode et al. (2016) showed that the shape of the feature is sensitive to particle shape.**
    As these sentences only refer to our simulations and explain the observed results, we find that the sentences are correct as they are.

13. **L230ff See 1)-4): The observed spectral fingerprint is not exploited fully to define of criteria of the classification. E.g. the spectral region around 960 cm$^{-1}$, which was shown to respond significantly to size distributions and particle shape (see Woiwode et al., 2016) is not considered. It is not clear to me whether modulation of the spectra by scattered light from the below is considered. Direct comparisons between observed and simulated spectra are not presented to verify the underlying assumptions.**
    See general comments 1-4.

14. **L279, L314, L336, L507 It should be noted that these statements on the performance are valid within the assumptions made. Verification by direct comparisons of simulated and observed spectra and comparisons with other observations are not provided here.**
    As the whole Section 3 describes our simulations and the derived results, the statements are given with respect to the simulations. Nonetheless, we added some additional information on that.
    In L507 the statement is made with respect to the observations, but as we showed spectra showing the shifted feature and, additionally, added comparisons between simulations and observations (see general comments 1), we think the statement is correct here.

15. **L315ff For comparisons with other studies, it would be helpful to include a Figure showing the used size distributions.**

As the number of scenarios of bi-modal NAT particle size distributions is 115, the number of size distributions is also. We think that this is too much to show all size distributions. Furthermore, all relevant parameters of the size distributions are given in Tab. 3.

16. **L383ff See 6): It would be interesting if the authors could discuss how the method responds if the cloud bottom is located above flight altitude and in the presence of several PSC layers located above each other. Such conditions are frequently found inpolar winters (e.g. Pitts et al., 2018).**
See general comments 6 and section 3.5.

17. **L392 The motivation for choosing the discussed flight should be provided.**
We also included results for other flights (see general comments 1). We show more details for flight 3 as it shows the largest NAT signal of all flights and gives a good example. This information is added.

18. **L397 It would be helpful to indicate the location of the selected two profiles in Fig. 10 and comment on the composition derived in Fig. 10b.**
We added arrows to the Fig 10 a) and b), which are now Fig. 11 b) and c).

19. **L416ff See 5): Considering one case study constitutes only a limited test for a new classification scheme. Further RECONCILE PSC flights, collocated in situ observations and e.g. CALIPSO observations could be used to test (and potentially optimise) the proposed classification scheme.**
We added additional flights (see general comments 1).

20. **L422 Why are only spectra with CI < 3.0 shown here? (compare Fig. 3 and L388)**
The separation between the different types of PSCs and also for the different NAT sizes diminishes with increasing CI. This can be seen in Fig. 4. When all spectra with CI < 5 are considered more of the large particles cannot be detected and also medium size particles are missing. Furthermore, the separation between the different size regimes is not so distinct. Thus it is advisable to use only spectra with CI < 3, as these problems are largely reduced then. We added information to Sect. 3.2.1.

21. **L424 See 1): Figure 11 shows examples of PSC spectra in radiance units, while Figures 1 and 5 show simulations in arbitrary units, which are individually scaled. Clear comparisons between simulations and observations are not possible. Only a part of the used spectral range is shown for the observations (compare L99). Overlays or residuals between simulated and observed spectra using the same units would be helpful to evaluate to which degree the simulations meet details of the observations.**
See general comment 1.

22. **L435,436 See 5): Here, the CRISTA-NF observations are classified as STS or medium sized (median r=1.5-4.0$\mu$m) NAT. However, collocated in situ observations suggest particles with radii exceeding 7.5 and 10$\mu$m (see Molleker et al., 2014, Table 1). Are these results compatible? It would be interesting to compare size distributions, which are supported by the CRISTA-NF**

**simulations, with the in situ observations.**
See general comment 5.

23. **L472 The conclusion on the spectral range from 833 to 960 cm$^{-1}$ is not supported sufficiently by Fig. 5, since the spectral range from 840 to 940 cm-1 is not shown.**
In our opinion it is supported. Both spectral ranges are shown and there is no difference between the two spectral ranges in relative as well as in absolute values. Nonetheless, we rephrased the sentence and used the term difference.

24. **L474ff From my point of view, the conclusion "very similar" is not sufficiently supported. The Woiwode et al. (2016,2019) studies used a broader spectral range, higher spectral resolution and spectra in absolute radiance units (which constitutes another "piece of information"). Furthermore, these studies used direct overlays and residuals of simulated observed spectra to analyse the spectral fingerprint of large NAT particles in detail and to define criteria for detection. In the study presented here, the full available spectral range is not exploited and no direct comparisons between simulated and observed spectra are provided. However, within the above mentioned limitations, the conclusion on similarity is supported to some degree by the Woiwode et al. (2016) study, Appendix B, for the small and medium particle sizes. In this study it was shown that for small NAT particles (r=1$\mu$m), spherical and aspherical particle populations show an almost identical spectrum, while at r=3$\mu$m the signatures start to diverge notably. Furthermore, one might speculate that particles become gradually more aspheric when growing to large sizes, and that (nearly) spherical particles are a suitable assumption at earlier growing stages.**
It is correct that Woiwode et al. (2019) used a wider spectral range, a better resolution etc. for their simulations. Thus, a direct comparison between the two simulations is not advisable. However, as the simulation for the bimodal NAT size distribution shows an enhanced step from 818 to 833 cm$^{-1}$ and no difference between the spectral regions 833 to 960 cm$^{-1}$, in our opinion the spectrum will fullfil all criteria of the described detection method. For this "comparison" the whole spectrum is not necessary and we simulated all spectral windows used by Woiwode et al. (2019) algorithm. Unfortunately, a direct use of this method is not possible as the absolute radiance values of the two instruments, CRISTA-NF and MIPAS-Envisat, are different (see specific comments 27.).
We rephrased the sentence accordingly.

25. **L476ff It has been shown before that a similar signature can be simulated by assuming spherical particles (see Woiwode et al., 2016, Appendix B). Furthermore it should be noted that the possibility of some variability in the NAT phase regarding particle habits has been clearly mentioned in previous studies (Molleker et al., 2014; Woiwode et al.,2016) and has not been ruled out in the Woiwode et al. (2019) study. However, using the combination of a wide spectral range, high spectral resolution, and supporting information from in situ observations, the Woiwode et al. (2016,2019) studies were not successful in reproducing details of developed spectral fingerprints of large NAT particles by assuming spherical particles. The observed combination of**

a strong "step-like" feature and a flat spectral baseline of the observed spectra towards higher wave numbers (described by the simplified "hockey-stick" picture in Woiwode et al.,2019) could not be reproduced. Close inspection of the spectra showed that Mie simulations of spherical particles always showed significant differences from the observed "shifted feature" around 820 cm$^{-1}$ and a significant negative slope and/or an "upwardarching" of the spectral baseline at higher wave numbers, and further discrepancies. These discrepancies resulted in significant patterns in the residuals. Using spectral characteristics of highly aspherical particles clearly improved the residuals. Thus, the spectrum clearly includes information on particle shape. Also from other work it is known that infrared spectroscopy allows to infer information on particle shape (Wagner et al., 2005). However, it is understandable to me that within the limited spectral range considered here and without detailed comparisons between observations and simulations a clear decision appears to be not feasible here. From my point of view, the question is not only whether any "hockey-stick" signature can be modelled, but also whether details of the entire accessible spectral fingerprint in developed signatures of large NAT particles can be reproduced, and how the results compare with other observations.

Although our simulations as well as the CRISTA-NF measurements include the three spectral windows, here we consider our 950 cm$^{-1}$ a sufficiently good substitute for the MIPAS 960 cm$^{-1}$ window, proposed by Woiwode et al. (2019) to identify aspherical NAT, we cannot apply the method as it relies on absolute radiances that are inherently different for space-based MIPAS and air-borne CRISTA-NF measurements. However, assuming spherical NAT particles and a bi-modal size distribution our simulated spectra qualitatively reproduce the spectral signature attributed to aspherical particles. To our knowledge Woiwode et al. (2019) did not investigate the effect of bi-modal size distributions on the spectra and hence, we think this should not be ruled out. We rephrased the sentence here.

26. **L477f This sentence should be revisited, since this has been done in before (Woiwode et al., 2016,2019). Of course, any further case studies including in situ comparisons would be helpful to further constrain properties of large NAT particles.**
Simulations for single spectra are shown, which are in agreement with the FSSPs and the situation. We added a reference. But probably more additional information is necessary as the FSSPs only measure at flight altitude and an infrared limb sounder observes a large volume from flight alitude up to a few hundred kilometres away.

27. **L480f Here, clarification is required. Since absolute radiances are shown in Figure 11, I would expect that integrated radiances and their differences can be calculated.**
As MIPAS-Envisat is a satellite instrument and CRISTA-NF an airborne instrument and both have significantly different viewing geometries, spectral resolution, spectral sampling, field-of-view, etc., the absolute radiance values are not the same, even when the same air masses have been observed. Thus, also the differences are not the same and the method cannot be applied. We rephrased the sentence for clarification.

28. **L484 See comment to L476ff: it has been shown before that a similar signature can be simulated by assuming spherical particles. However, close inspection showed significant discrepancies between simulated and observed spectra in the case of spherical particles.** Please see answer to comment 25.

29. **Fig 11 The channel LRS 5 seems not to be shown (compare L99).**
We extended the figure and now show also the region around 950 cm$^{-1}$ in the way as for the simulations.

30. **Fig 10 An additional panel including a map with the geolocations of the observations would be helpful to visualize the location and size of the sampled region.**
We added this panel.

**Technical corrections**

1. **L32 denitrifaction -> denitrification**
corrected

2. **L115 KOPRA should be expanded**
done

3. **L450 CALIPSO and CALIOP should be expanded and a reference should be provided (e.g. Pitts et al., 2018)**
done

4. **L458 transfered -> transferred**
done

5. **L478 FSSP should be expanded and a reference should be provided**
done

6. **L500 GLORIA should be expanded and a reference should be provided**
done

7. **L508 Sentence should be revisited. How can new PSC observations be obtained from the presented method? (Possibly, "observations" should be replaced e.g. by "data set")**
In the preprint version the sentence is: "... new data set of PSC observations ..."

8. **Fig 3, Fig 6 Plots may be refined: data points seem to overlap strongly and it is not clear whether significant populations of data points are hidden below other data points. At least, this should be discussed in the text.**
We added information to the text.

9. **Caption Fig 4 spetrum -> spectrum** done

10. **Fig 7b The x-axis might be scaled from e.g. -4 to >8 for a better focus on the relevant region** We changed the scale to -4 − 8. As the gradient exceeds 20 only at one point we avoid to show the full positive range to show more details in the other range.

---

## Author Comment (AC2) · 26 Oct 2020

Reply to the comments of reviewer 2 on the manuscript

Radiative transfer simulations and observations of infrared spectra in the presence of polar stratospheric clouds: Detection and discrimination of cloud types

by C.Kalicinsky et al.

We thank the reviewer for the helpful comments and recommendations. In the following, we discuss the issues addressed by the reviewers and explain our opinions and the modifications of our manuscript.
We enumerate the comments and repeat them in bold face. The modifications of the manuscript are displayed in the marked-up manuscript version as colored text. Deleted parts are shown in red and new or modified text parts in blue.

**1 Comments**

**The paper describes methods to detect PSCs and to classify them. The following PSC types are distinguished: NAT (nitric acid trihydtrate), STS (supercooled ternary solution) and ice clouds. For the detection of small NAT particles, a spectral feature at about 820 cm$^{-1}$ has often been used. The authors show, that for larger NAT particles this feature shifts towards smaller wavenumbers and they use this feature to further specify the NAT particles as sNAT (small particles), mNAT (medium particles), and lNAT (large particles). Ice is detected by using the difference between 2 spectral regions ($\sim$833 cm$^{-1}$ and $\sim$950 cm$^{-1}$), the radiance is similar in clear sky conditions but significantly smaller at 950 cm$^{-1}$ when ice is present, a well-known feature which has already been used before. When the cloud is neither NAT nor ice, it is classified as STS. Further a method based on the so-called Cloud Index (CI) is developed to derive the cloud bottom height.**
**The methods are applied to a comprehensive set of radiative transfer simulations including NAT, ice and STS, all with various concentrations and particle size distributions. Based on the simulations it is demonstrated that the methods work well, in particular sNAT can be distinguished very well from the larger NAT particles mNAT and lNAT. Finally the new methods are also applied to observations of the CRISTA-NF instrument taken during a flight of the RECONCILE campaign**
**Generally the newly developed methods are described quite well and the figures are appropriate. However, I think that some clarifications are needed, in particular the motivation is not so clear for readers who do not work in the specific field of research (see also my comments below). The scope of the paper fits AMT very well, therefore I recommend publication after taking into account the comments below.**
**General comments:**

1. **The motivation is not clear. It is said in the abstract that there are "uncertainties in the representation of PSCs in model simulations". Which models? Atmospheric chemistry models, climate models? Please provide**

**some examples of models which can be improved by better knowledge of PSC properties. Which properties are required by these models? What are the most relevant parameters? Composition, concentration, size, shape ...? The methods presented here do not derive number concentrations, can these also be derived from spectral IR observations? How important is the classification into different size regimes?**

Especially, chemistry climate models (CCMs) that are used to asses polar stratospheric ozone loss (e.g. Eyring et al. (2013)) often use rather simple schemes to represent PSCs in the model simulations. Such simplifications may lead to a heterogeneous chemistry dominated by NAT, but it is known that heterogeneous chemistry on STS and cold binary aerosol particles probably dominates the chlorine activation (e.g. Solomon (1999), Drdla and Müller (2012), Kirner et al. (2015)). Additionally, no comprehensive microphysical models are typically used to describe evolution of PSCs over the winter. Mesoscale temperature variations that are known to play an important role for the formation of PSCs (Carslaw et al. (1998), Dörnbrack et al. (2002), Engel et al. (2013), Hoffmann et al. (2017)) are also missing in current state of the art CCMs (Orr et al. (2015)).

Assumptions on the occurence of different PSC types typically have only limited impact on many aspects of ozone loss, as, for example, liquid PSC particles are sufficient to simulate nearly all ozone loss (Wohltmann et al. (2013), Kirner et al. (2015), Solomon et al. (2015)). However, there are also situations where the PSC type is crucial. For example, which PSC type is present at top of ozone loss region is important (Kirner et al. (2015)) and during the initial activation in PSCs covering only a small part of the vortex the type plays also an important role (Wegner et al. (2012)).

Furthermore, the heterogenous reaction rates on PSCs strongly depend on temperature but also on the PSC type (e.g. Drdla and Müller (2012), Wegner et al. (2012)). Here, especially for NAT the reaction rates show rather large uncertainties (Carslaw et al. (1997), Wegner et al. (2012)), which highlights the importance of observing the PSC type.

In summary, information on the compostion of the PSCs is very important, but measurements are limited. Typical PSC measurement techniques are in-situ particle measurements and lidar observations (e.g. Molleker et al. (2014), Achtert and Tesche (2014), Pitts et al. (2018)). But beside these measurement techniques an infrared limb emission sounder builds a good basis for such kind of studies.

We added these information to the introduction.

2. **Cloud optical thickness is a useful parameter to describe a cloud, the term is sometimes used in the paper but no number is given. What is the optical thickness range of the clouds considered here (in the RT simulations and what can be observed using the limb observations)? I suppose that all clouds are optically very thin, what is the upper limit of cloud optical thickness that can be analysed with the presented methods?**

We added the extinction ranges to Tab. 3 for each individual PSC type. They range from $1.0e^{-5} - 1.0e^{-2}$ km$^{-1}$, whereby the largest extinction is only achieved by ice. For the optical thickness (= extinction * vertical thickness) the range is then $0.5e^{-5}$ to $8.0e^{-2}$ for ice, from $1.05e^{-5}$ to $3.3e^{-2}$ for STS and from $1.7e^{-5}$ to $1.76e^{-2}$ for NAT.

As the clouds are typically detected using the CI and a threshold value (here 3.0), the

upper limit with respect to the optical thickness cannot be stated here. As the CI depends on many different factors (e.g. cloud type, particle radius, altitude of the cloud etc.), the CI values for the same extinction and optical thickness can be different (see General comments Reviwer 3). Thus, the upper limit also depends on these factors influencing the CI and cannot be stated in general.

3. **The radiative transfer simulations are based on a single scattering approach which is not described in detail here. This also means that it is only valid for very thin clouds, for which multiple scattering can be neglected (see e.g. Höpfner and Emde 2005). Please provide the optical thickness range used for the radiative transfer simulations to justify the neglect of multiple scattering.**
We compared our results with the findings by Höpfner and Emde (2005) and estimated the maximum uncertainties that can occur. In the case of STS the SSA of 0.24 also fits to our simulations. The scenarios defined in Höpfner and Emde (2005) that fit to our simulations are typically 1 and 2, maybe for a small part of the simulations it is scenario 3. This leads to uncertainties typically $\leq 1\%$ (4.5% for some simulations). In the case of NAT the scenarios are 1 and 2. The SSA lies between the two SSA simulated by Höpfner and Emde (2005) of 0.24 and 0.84. Here we took a mean SSA of 0.54 and a mean uncertainty of the both analysed SSAs. Then for NAT the uncertainty is $\leq 4\%$. For ice the scenarios are mainly 1 and 2 and for a small portion of the simulations (those with the largest volume density) also scenario 3 fits. The SSA is comparable to NAT and we used 0.54. This leads to uncertainties $\leq 4\%$ ($\leq 20\%$ in case of sceanrio 3). In total for almost all simulations the uncertainties are $\leq 4\%$ and only a few simulations have larger uncertainties. But with respect to the computational effort these small uncertainties do not justify the use of multiple scattering.
Furthermore, in our analysis we typically use radiance ratios. As the single scattering approach leads to an underestimation of the radiance that is often similar in many spectral regions, the uncertainties of the ratios are much smaller than the uncertainty of the radiances themselves. E.g. an underestimation of the radiance by 10% in all spectral regions would lead to the same ratios.
We added information to the Sect. 2.2.

4. **Presuambely, the JURASSIC model does not account for horizontal inhomogeneities. Please discuss the validity of this approach for PSCs, i.e. how large is the horizontal extend of the PSCs typically compared to the line of sight through the PSCs?**
Our simulation setup does not account for horizontal inhomogeneities of the PSC. The horizontal extent of the line of sight of the instrument inside the PSC can reach up to several hundred kilometres. In case of synoptic scale PSCs horizontal homogeneity is a good approximation. Other events, such as e.g. mountain wave ice, can lead to PSCs with a smaller horizontal extent. But with respect to the large amount of PSCs simulated in this study we started with the most simple approach regarding the horizontal homogeneity of the PSCs.
We added some information to Sect. 2.2.

5. **For the reader it is rather difficult to remember the definition of all indices used in the paper. I think it would be helpful to include a figure showing**

**the spectral regions used and mark the spectral windows that are used to calculate the individual indices. Also a table including the definitions of NAT-indices 1,2,3 and the CI index could be useful.**

We inserted the different regions into Fig. 1 and numbered the micro windows (MW) from 1 to 7. We additionally added a new table for all indices, where the ratios and BTD that are used can be seen.

6. **I miss some discussion about the uncertainties. For example on p.8 you write: "Nearly all simulations with NAT particles $< 3\mu$m lie above the region of the simulations for STS and ice clouds, which is marked by the solid black line. Thus, NAT particles within this size range can be detected and discriminated using NAT index-1." For the modelled spectra this is correct, but also for real observations? Measurements always include uncertainty, how accurate measurements are required so that NAT is clearly separated from the ice clouds?**

Because of the strong radiance enhancement caused by the clouds, the dominating uncertainty is the relative uncertainty (noise uncertainty) of the measurements, whereas a systematic uncertainty (e.g. an offset) plays a minor role. The relative uncertainty for the radiances typically observed during atmospheric measurements is about 1–2% for CRISTA-NF (Schröder et al. (2009)). Because of the calculation of ratios, the relative error can add up but is still only a few percent. In the worst case (one MW plus 1-2% and the other MW minus 1-2%) the maximum uncertainty is about 2-4%. For the analysis of the PSC observations this has a different effect depending on the CI and the NAT-index. For smaller values of these quantities the resulting absolute uncertainties of the quantities determined from the relative uncertanties of maximum 4% are smaller than for larger values, e.g. a CI of 2 has a maximum uncertainty of $\pm$ 0.08 and a CI of 5 an uncertainty of $\pm$ 0.2. Thus, the methods work better for smaller CI values. This suggests to use a threshold value for the CI and to only analyse observations with a CI below this threshold. This suggestion we consider supported by our finding that the separation between the different size bins was better for a CI below 3 (Fig. 3 and Fig. 4). So we keep this value. Furthermore, the same spectral windows are used for CI and NAT index-1/2. In this case some of the uncertainty will cancel, as the data points will shift similar to the correlation regions/lines, i.e. a smaller CI (because of a smaller randiance in the $CO_2$-window) will be accompanied by a larger NAT index and vice versa. Lastly, the scatter plots for the observations (new Fig. 9), especially for flight 3 and 5, show that a large number of observations exhibit deviations from the separation line larger than a few percent and a clear separation between ice (flight 1) and NAT is visible in Fig. 9 a) and b). Furthermore, the measurements show a compact correlation, which we consider an indications for little noise, whereas we would expect more spread for large noise errors.

Information on the uncertainties are added to the text.

**Specific comments:**

1. **The title is a bit misleading. The main focus of the paper are the methods to classify and detect PSCs. I thought from reading the title starting with "Radiative transfer" that the paper was more about radiative transfer methods etc. For the RT simulations a well-known model JURASSIC is applied**

**but the methodology is not described in this paper. Also the observational methods are not described here. I think that the title should be something like e.g. "A new method to detect and classify polar stratospheric clouds"**
We changed the title to:
A new method to detect and classify polar stratospheric NAT clouds derived from radiative transfer simulations and its first application to airborne IR limb emission observations

2. **Abstract: p1, l7 "... showed a spectral peak at about 816 cm-1 . This peak is shifted compared to the peak at about 820 cm-1, which is known to be caused by small NAT particles. " -> A bit more information about this peak would be helpful. What is the physical process responsible for the peak. Which physical processes could produce a shift of a spectral peak ...**
The peak at $820^{-1}$ is mainly caused by emission of radiation. The transformation then is caused by the increasing contribution of scattering to the total extinction. We briefly added this to the abstract.

3. **p1, l16: "gradient of the CI" -> which gradient is meant here? -> "vertical gradient"**
yes, we corrected this

4. **p3, l31-33: These 3 sentences should be shifted to Section 2.2**
These 3 sentences are only a short summary was is comming in the long Sect. 2 in total. The details are then given in the subsections and Sect. 2.2 describes the radiative transfer code JURASSIC. We find that these sentences do not fit to Sect. 2.2. Thus, we stay with the old text.

5. **p4. l19: Title of section should include the term "radiative transfer simulations"**
we changed the title to "Radiative transfer simulation code JURASSIC"

6. **p4 l32: "The optical properties of the particles, extinction coefficient, scattering coefficient, and phase function, required for the radiative transfer simulations ..." -> "The optical properties of the particles (extinction coefficient, scattering coefficient, and phase function) required for the radiative transfer simulations ...", include brackets here because extinction coefficient, scattering coefficient, and phase function are the optical properties**
done

7. **p7 l20: What is the "scattering radius" of a particle, this term has not been defined**
The scattering behaviour of a PSD depends on its median radius $\mu$ and distribution width $\sigma$ and the wavelength. In many cases the effective radius of a PSD is a sufficiently good approximation to describe the scattering behaviour with a single parameter. However, when particle size and wavelength are approximately the same size, this is not a good approximation as the scattering behaviour for two PSDs with same $r_{eff}$ but different $\mu$ and $\sigma$ can be different. Here, the scattering radius $r_{sca}$ is a better single

parameter to describe the scattering behavior:

$$r_{eff} = \frac{\int_{r_0}^{r_1} \pi r^3 n(r) \mathrm{d}r}{\int_{r_0}^{r_1} \pi r^2 n(r) \mathrm{d}r} \tag{1}$$

$$r_{sca} = \frac{\int_{r_0}^{r_1} \pi r^3 n(r) Q_{sca} \mathrm{d}r}{\int_{r_0}^{r_1} \pi r^2 n(r) Q_{sca} \mathrm{d}r}, \tag{2}$$

where $Q_{sca}$ is the scattering efficiency depending on radius $r$, wavelength and complex refractive index e.g. see Hansen & Travis (1974). Since we assumed only one $\sigma$ in our simulations the PSD median radius is sufficient to unambiguously characterise the scattering behaviour in our study.

We revised the sentence:
" The appearance of the spectral feature that is observed in infrared limb spectra in the presence of polar stratospheric clouds consisting of NAT particles in our study is unambiguously characterized by the median radius of the particle size distribution, as we kept the distribution width $\sigma$ constant."

8. **p7 l31: "different contributions of extinction and scattering" -> extinction is the sum of absorption and scattering, therefore I think that you mean "absorption and scattering"**
yes, we corrected this

9. **Fig.1 and 5: "spectra have been scaled such that the radiance equals 1" -> scaling factor is not clear, is it the average radiance over the plotted spectral range?**
The scaling factor is 1/(mean radiance in window 832-834m$^{-1}$). We added this.

10. **Fig.2: For the interpretation of the RT simulations, it would help to include here also the refractive indices of ice and STS. Further I think that it would be very helpful to show extinction and absorption coefficients, which are probably calculated using Mie theory for the individual PSC types and for some particle sizes to see, that for larger particles the scattering coefficient dominates.**
We added the refractive indices for STS and ice. Furthermore, we show two more plots now, the extinction and the SSA for NAT with different median radii. These plots illustrate the behaviour of the radiance spectra with increasing median radius. We rephrased the text in Sect. 3.1 accordingly.

11. **p9, l4: "results for PSCs with larger NAT particles (up to 4$\mu$m) also lie above the simulations for STS and ice (black separation line)": In Fig.3 it is not well visible, which radii lie above the separation line. May be discrete colours could be used for each of the simulated radii?**
We used new discrete colours for the Fig. 3. Additionally, we also changed the colours in Fig. 1, Fig. 2, and Fig. 6 to use always the same colours for the NAT, STS, and ice simulations.

12. **p9, l23: "detected NAT spectra" -> NAT is detected, not the "spectra"**
we rephrased the sentence

13. **p10, l23: "When the NAT detection and classification procedure (described in the previous subsection) is applied to the simulation results for the mixed clouds, the good discrimination between the small and medium size particles remains." -> this is not shown, why?**

The plot is limited to the range 0.5 to $3.5\mu$m as only a subset of the NAT simulations is combined with STS. Thus, we decided to give the most important numbers regarding the detection capacity in the text and show the plot here in the reply.

[Figure]

14. **p13, l3-9: Here you discuss about "cloud optical thickness" (thick and thin) without providing any values -> as mentioned above, please quantify the cloud optical thickness**

The terms thin and thick in this context are relative terms. Unfortunately, a clear relationship between the optical thickness and the CI does not exist (see General comments Reviewer 3). As all different clouds (different type, radius, altiude etc.) enter the analysis here the optical thicknesses can be different for the same CI values.

Nonetheless, we rephrased the text to clarify the meaning. There are two cases where a bottom altitude cannot be detected. In some cases the observations run into saturation and the CI values below the cloud bottom altitude stay very low. Additionally, in some other situations the CI values below the bottom altitude only show a linear increase and not the larger change directly below the bottom altitude. In both cases the approach fails. A way to sort out such possibly affected observations is the use of a threshold value. For our simulations a CI minimum of 1.25 turned out to be sufficient. Such low values of CI only occur for a part of the ice clouds simulated here (typically that with the largest volume densities and for a few of the NAT and STS clouds with large $HNO_3$ VMRs ($\geq$ 11 ppbv) or volume densities ($\geq$ 5 $\mu$ m$^3$/cm$^3$) combined with a large vertical thickness ($\geq$ 4 km).

15. **p15, l7: "... where much more NAT was observed by CALIOP (the difference in the Southern hemisphere is much smaller)"-> how much more NAT was observed by CALIOP, how much smaller is the difference in the Southern hemisphere?**

The agreement for NAT in the Southern hemisphere is 73%, whereas there is only an agreement of 18% in Northern hemisphere. We added these numbers to the text.

16. **p16 l20: "This method can surely be transferred to other cloud observations**

**such as cirrus clouds and aerosol layers and to other airborne instruments measuring in the same wavelength region like e.g. GLORIA." -> Which clouds and aerosols could be observed? Probably only very thin clouds? Up to which cloud optical thickness can the method be applied?**

As said before, our method rely on the CI and there is no clear relationship between the CI and the optical thickness (see General comments Reviewer 3). Thus, a clear statement cannot be made as the upper limit depends on many factors.

---

## Author Comment (AC3) · 26 Oct 2020

Reply to the comments of reviewer 3 on the manuscript

Radiative transfer simulations and observations of infrared spectra in the presence of polar stratospheric clouds: Detection and discrimination of cloud types

by C.Kalicinsky et al.

We thank the reviewer for the helpful comments and recommendations. In the following, we discuss the issues addressed by the reviewers and explain our opinions and the modifications of our manuscript.
We enumerate the comments and repeat them in bold face. The modifications of the manuscript are displayed in the marked-up manuscript version as colored text. Deleted parts are shown in red and new or modified text parts in blue.

**1 Comments**

**This paper demonstrates the clear capability of infrared FTS limb sounders to provide detection, discrimination of particle types and particle sizing in polar stratospheric clouds and is an advance on the current state of the art. The paper is acceptable for publication following some minor corrections.**
**General comments:**

1. **I strongly suggest that an attempt is made to make an additional plot that shows the optical depth vs CI for some samples of different PSC cloud types. Likewise the CI vertical gradient is related to the optical thickness gradient.** Unfortunately, there is no clear relationship between the optical depth and the CI. Different parameters can influence the CI that leads to different values although the optical depth is the same.
Spang et al. (2008) already showed that the CI depends on the altitude and that additionally the background atmosphere (e.g. polar winter vs. tropics) can have a large influence. Griessbach et al. (2014) showed that the observed radiance, and thus the CI, also depends on the radius of the particles. In Griessbach et al. (2020) the authors showed that there is some correlation between CI and extinction for ice and volcanic aerosol but a distinct relationship could not be determined. CIs of large particles (r > 5 $\mu$m) are more related / show a quite good correlation with the integrated surface area densities along the line of sight or for ice with the ice water path divided by effective radius (Spang et al. (2012, 2015)). We also did some studies with the new simulations and found similar results. Furthermore, the particle type also plays a role as the spectral slope of the extinction is different for the different particle types. Thus, the radiance enhancement in the regions used for the CI can be different although the total extinction and thus the optical depth are the same. As a consequence the CI values can only be related to an optical depth when all influencing parameters (altitude and thickness of cloud, particle type and radius, background atmosphere) are known, which is typically not the case. Therefore, we cannot give numbers in this direction in the paper as there are too many unknowns.

**Specific comments and typos**

1. **Page 2 L35: "incomplete" rather than "difficult"?**
   changed the term

2. **L37-38: [An] infrared .... build[s]...**
   done

3. **L39: sounder[s]**
   done

4. **L44-45: make distinction between CO2 molecular line emssion and broader continuum like aerosol emission?**
   we rephrased the sentence

5. **L51: color => colour**
   done

6. **Page 3 L64-65: Maybe make clearer that for an airborne instrument the limb tangent moves away from the aircraft for downward looking views.**
   we added information

7. **L72: What about changes with the aspect ration of the particles?**
   We added information that it also depends on shape and radiation scattered from below. See Reviewer 1.

8. **L86: itselves => themselves**
   done

9. **Page 4 L87: JURASSIC is not defined until L106**
   we added the definition

10. **L110: PREMIER IRLS is not defined**
    we added the definition

11. **L111: spectral/ly/**
    done

12. **L115: KOPRA is not defined**
    done

13. **Page 5 L130: What about spectral regions for STS and background binary sulfate aerosols?**
    As STS and the background binary sulfate aerosol have no distinct spectral features like a peak nor a clear slope of the extinction such as ice, there are no special regions for STS and the background aerosol.

14. **L133: al[t]itude**
    done

15. **L139: reference to a rejected ACPD paper?**
    The paper has not been rejected, but the reply to the reviewer comments and the upload of a revised manuscript have not been done. However, it is the best paper describing the atmosphere and it has been cited far more than 100 times.

16. **Page 6 L177: median radius varied in steps of?**
For small particles it has been varied in steps of 0.5 $\mu$m, then in steps of 1 $\mu$m, and at the end there is one step of 2 $\mu$m. Because of the different step sizes all used radii are summarised in Tab. 3.

17. **Page 8 L215: imaginary part/s/**
done

18. **L234-249 and everywhere else including figure captions and tables: Is it possible to give all these spectral regions a distinct short name? e.g. R1, R2, R3 etc Otherwise the reader has to scan the characters and check to see which regions are the same thing rather than seeing that immediately from the short name.**
We added a table for all indices and we marked the regions in Fig. 1 (see Reviewer 2). Additionally, we numbered the micro windows (MW) from 1 to 7 and added the short names like MW1 at the corresponding text positions.

19. **Page 9 L257: an[d]**
done

20. **Page 10 L312: less => fewer**
done

21. **Page 11 L335: mille => thousand**
done

22. **Page 13 L408: extend => extent**
done

23. **Page 15 L463: called [a] hockey-stick**
done

24. **Page 16 L494: What are the detection levels of the new method compared to the old method? e.g. in terms of the minimum volume density um3/cm3.**
The detection level for the small particles is the same as for the old method, as they are detected with the same method. The improvement of our method is that the detection is expanded to larger NAT particle sizes that are not detectable with the old method.

25. **L501: "minimisation" means "reduction"?**
yes, we changed this

26. **L507: "safely" means "always"?**
we removed safely

27. **Figures 3, 6 and 7: Can an approximate optical depth scale be put on the x-axis?**
Unfortunately, there is no distinctive relationship between optical depth and CI (see General comments).

28. **Figure 8: What are the actual optical depths corresponding to these CI values?**

Unfortunately, there is no clear relationship between optical depth and CI (see General comments). As clouds with many different parameters enter this plot, the answer cannot be given.

---

## Author Response (AR2)

Reply to the reviewer comments on the manuscript

A new method to detect and classify polar stratospheric NAT clouds derived from radiative transfer simulations and its first application to airborne IR limb emission observations

by C.Kalicinsky et al.

We thank the reviewers for their helpful comments and recommendations. In the following, we discuss the issues addressed by the reviewers and explain our opinions and the modifications of our manuscript.
We enumerate the comments and repeat them in bold face. The modifications of the manuscript are displayed in the marked-up manuscript version as colored text. Deleted parts are shown in red and new or modified text parts in blue.

**1 Comments Reviewer 1**

**From my point of view, the authors clearly have improved the manuscript, and I appreciate their efforts very much. I would suggest to consider the following points prior to final publication. Indicated figure numbers and line numbers refer to the manuscript including track changes.**

1. **Depending on the scope of the paper, a more direct comparison of observed and simulated spectra would be interesting to show how to which degree the observations are reproduced from a spectroscopic point of view. However, if the goal is mainly to show that a signature similar to the observed NAT feature can be simulated qualitatively in the considered spectral range to support the proposed size classification, and a quantitative simulation is beyond the scope, the indirect comparison of Figs. 1,5, with Figs. 9f, 12 may be sufficient from my point of view. In this case, this aspect (i.e. best-possible qualitative approximation of the observed feature within underlying assumptions) should be mentioned more clearly in the sections 5 and 6.**

   As our analysis method is based on color ratios, i.e. relative differences, we can only compare observed and simulated spectra qualitatively. Thus, the presentation of the observed spectra in Figs. 9f and 12 are for illustration and it is not the intention of the paper to show direct comparisons.
   It is correct that the analysis method searches for spectra that qualitatively show the same features and, thus, appear in the scatter plots in the same regions. Therfore, it is a qualitative approximation and not a quantitative one. An advantage by using this relative approach is that differences in the absolute radiance values only lead to changes inside the correlation regions for a specific particle type or size.
   We added additional information to Sects. 4.2.2, 5., and 6..

2. **In the discussion section, it should be clarified that the presented qualitative simulations of the NAT feature do not constitute a proof that the spectra attributed to highly aspherical in the Woiwode et al. (2016, 2019) studies can be simulated quantitatively by the proposed method on the basis of spherical particles. From my point of view, the statements at L608-L613**

**are supported only qualitatively by the presented results and are limited to the considered spectral range used here. Therefore, I would suggest to clarify these aspects. Regarding L619, I cannot confirm that the shifted NAT feature in the previous study was solely attributed to asphericity (mode radius was also important, see Woiwode et al. (2016), Fig. 12, Fig. B2 and conclusions).**

As mentioned in 1. our method compares spectra qualitatively. Thus, it is correct that the different appearances qualitatively can be reproduced using spherical particles. It was not the intention to say that we definitely can quantitatively simulate the spectra shown in Woiwode et al. (2016, 2019) by only using spherical particles. We clarified this in the text.

Regarding L619, we changed the text and mentioned that the combination of the correct particle size distribution in combination with asphericity was important to simulate the observed spectra.

3. **I appreciate the information provided by the authors regarding the influence of tropospheric clouds and I agree that the presence/absence of tropospheric clouds will mainly contribute a broadband-component to the spectrum and weaken the amplitude of the NAT feature. Since the scattered tropospheric component of the spectrum also modulates the appearance of the NAT feature to some degree (compare refractive index real part in Fig. 2a), I would suggest to double-check or discuss in more detail whether this aspect could affect the conditions used for size classification.**

   We again used the simulations for the MIPAS-Envisat instrument (see e.g. Spang et al., 2012, 2016) to evaluate the effects of tropospheric clouds in more detail. These results can be transferred to the CRISTA-NF simulations.

   First, in the case of STS and ice no effect on the analysis can be seen as the spectra show no features in the range 810–820 $cm^{-1}$. The effect by the change of the absolute radiance values cancels because of the use of ratios, i.e. relative changes. Thus, the separation lines, which are defined using the STS and ice simulations, remain and no false detection of ice or STS by assigning these particle types to a NAT class can occur. Second, for NAT clouds the impact of underlying clouds slightly depends on the median radius of the particles, as the scattering contribution increases with radius (see Fig. 2 c)). For median radii $\leq 1.0$ $\mu$m an influence of a tropospheric cloud is negligible, as the spectra are dominated by absorption/emission. For larger median radii a little change of the features can occur and a little change of the positions of the data points in the scatter plots is possible. Possible effects eventually are an assignment of NAT with a median radius of e.g. 1.5 $\mu$m to the class sNAT (upward shift of data points for the difference NAT index-1 – NAT index-2) or a missing detection for lNAT (downward shift of data points for NAT index-3). Thus, a very little uncertainty of the class boundaries cannot completely be ruled out. However, the majority of all possible situations will still be detected and classified correctly.

   We added this more detailed information to the Sect. 3.2.2

4. **I would encourage to compare the collocated PSC particle observations from the same aircraft (Molleker et al., 2014, Grooß et al., 2014) at least briefly with the presented results in the manuscript.**

Grooß et al. (2014) presented a measured size distribution for a part of flight 3. The maximum in the distribution lies at a radius of about 2.5-3 $\mu$m and a second smaller maximum at about 5-6 $\mu$m is visible. Our results for this flight show mNAT particles (1.5 − 4 $\mu$m). This is in a reasonable agreement considering the limits of the comparisons. Note here, that the particle size distribution is observed solely at flight altitude and CRISTA-NF observes much larger air masses. Additionally, it is likely that the larger particles are more located at lower altitudes and the number density of these large particles will decrease with increasing altitude due to sedimentation. CRISTA-NF is also influenced by the air masses above flight altitude.

Molleker et al. (2014) presented particle size distributions for the flight 4 and 5. The maximum for flight 4 is at about 2.5-3 $\mu$m and additionally also larger particles were detected. Our results for this flight show mNAT, which is in agreement under the aforementioned restrictions for the comparison. For flight 5 different size distributions are presented, which show two maxima, one at about 2.5-3 $\mu$m and a second at about 5-6 $\mu$m. This flight shows the largest amount of large particles. Our results for flight 5 show mainly mNAT and a few lNAT. This is again in agreement within the scope to the limits of the comparison. But a detailed comparision would need additional information e.g. on the vertical change of the size distribution and the vertical extent of the cloud. We added these information to Sect. 5.

**2 Reviewer 2**

Technical corrections:

1. **l. 83 "where the tangent points move away from the aircraft with decreasing samping altitude" -> tangent points do not "move", please rephrase**
   We rephrased the sentence.

2. **l.159: "The observations were simulated in the altitude range from observer altitude ..."- > "tangent altitude range"**
   We rephrased the sentence.

3. **l.468: "... data points stay below the separation lines in the same way as the simulation results for ice" -> in the simulations part, it is said that the separiation line corresponds to the simulations for ice clouds, or the upper limit of the ice cloud simulations? (e.g. l. 278, "...ice clouds, which is marked by the solid red line". How exactly is the separation line determined for the analysis of simulations and observations? Please clarify.**
   The separation line is determined using the simulations of ice and STS to find an upper limit of these simulations. This is done in small CI bins, where in each bin the maximum NAT index value is determined and slightly shifted upward for safety. The separation line in the figure showing the observations is surely the same line, which was derived from the simulations. Thus, the fact that the observed results showing ice exhibit the same behaviour as the ice simulation results give additional confidence in the simulations and the correct choice of the separation line.
   We rephrased the corresponding sentences to clarify this.

4. **l. 517: "Thus, for most profiles a good estimate of the bottom altitude of the cloud is achieved" -> Could this be confirmed by independent observations?** Unfortunately not. No data or publications showing the exact bottom altitude of the cloud as it has been observed by CRISTA-NF a few hundred kilometres away from the aircraft are available to us. The were measurements of a downward looking lidar system carried out aboard the aircraft during the same flight, but to our knowledge these data have not been published and they are not available to us. Furthermore, a direct comparison would be difficult because of the different viewing geometries of the instruments.

[revised manuscript text omitted]